# Microbiome of Nodules and Roots of Soybean and Common Bean: Searching for Differences Associated with Contrasting Performances in Symbiotic Nitrogen Fixation

**DOI:** 10.3390/ijms231912035

**Published:** 2022-10-10

**Authors:** Flávia Raquel Bender, Leonardo Cardoso Alves, João Fernando Marques da Silva, Renan Augusto Ribeiro, Giuliano Pauli, Marco Antonio Nogueira, Mariangela Hungria

**Affiliations:** 1Department of Biotechnology, Universidade Estadual de Londrina, C.P. 10011, Londrina 86057-970, PR, Brazil; 2Soil Biotechnology Laboratory, Embrapa Soja, C.P. 4006, Londrina 86085-981, PR, Brazil; 3SuperBac Biotechnology Solutions, Estr. Vitória de São Pedro, 685, Mandaguari 86975-000, PR, Brazil; 4CNPq, SHIS QI 1 Conjunto B, Blocos A, B, C e D, Lago Sul, Brasília 71605-001, DF, Brazil

**Keywords:** symbiosis, inoculation, co-inoculation, *Bradyrhizobium*, *Rhizobium*, *Azospirillum*

## Abstract

Biological nitrogen fixation (BNF) is a key process for the N input in agriculture, with outstanding economic and environmental benefits from the replacement of chemical fertilizers. However, not all symbioses are equally effective in fixing N_2_, and a major example relies on the high contribution associated with the soybean (*Glycine max)*, contrasting with the low rates reported with the common bean (*Phaseolus vulgaris*) crop worldwide. Understanding these differences represents a major challenge that can help to design strategies to increase the contribution of BNF, and next-generation sequencing (NGS) analyses of the nodule and root microbiomes may bring new insights to explain differential symbiotic performances. In this study, three treatments evaluated in non-sterile soil conditions were investigated in both legumes: (i) non-inoculated control; (ii) inoculated with host-compatible rhizobia; and (iii) co-inoculated with host-compatible rhizobia and *Azospirillum brasilense*. In the more efficient and specific symbiosis with soybean, *Bradyrhizobium* presented a high abundance in nodules, with further increases with inoculation. Contrarily, the abundance of the main *Rhizobium* symbiont was lower in common bean nodules and did not increase with inoculation, which may explain the often-reported lack of response of this legume to inoculation with elite strains. Co-inoculation with *Azospirillum* decreased the abundance of the host-compatible rhizobia in nodules, probably because of competitiveness among the species at the rhizosphere, but increased in root microbiomes. The results showed that several other bacteria compose the nodule microbiomes of both legumes, including nitrogen-fixing, growth-promoters, and biocontrol agents, whose contribution to plant growth deserves further investigation. Several genera of bacteria were detected in root microbiomes, and this microbial community might contribute to plant growth through a variety of microbial processes. However, massive inoculation with elite strains should be better investigated, as it may affect the root microbiome, verified by both relative abundance and diversity indices, that might impact the contribution of microbial processes to plant growth.

## 1. Introduction

Some prokaryotes can catalyze the conversion of atmospheric nitrogen (N_2_) into ammonia (NH_3_), a process denominated as biological nitrogen fixation (BNF), which is key for the introduction of nitrogen (N) into natural and agricultural ecosystems [1]. The most successful contribution of BNF occurs in the symbiosis of bacteria collectively called “rhizobia” and plants of the Fabaceae family (syn. Leguminosae), encompassing legumes of great economic and environmental importance, including grain crops, green manures, forages, and trees [2]. Estimates are that in 2019, considering only the four main legume grain crops, around 34 million tons of N were globally fixed [3].

Rising prices of N-fertilizers and concerns about their impact on greenhouse gases (GHG) emissions and water pollution are driving quests to improve BNF’s contribution to agriculture. Interestingly, not all symbioses are equally effective in fixing N_2_, and understanding these differences represents a major challenge that can help in designing strategies to increase the contribution of BNF.

A contrasting example of the contribution of BNF relies on the comparison of the symbioses with soybean (*Glycine max* (L.) Merr.) and common bean (*Phaseolus vulgaris* L.). Soybean has a close symbiotic relationship with a few strains, majorly belonging to the species *Bradyrhizobium japonicum*, *Bradyrhizobium elkanii*, and *Bradyrhizobium diazoefficiens* and may reach outstanding contribution of BNF; it is the main crop using microbial inoculants worldwide, and Brazil takes the global leadership in N inputs with the legume [4,5]. In contrast, the contribution of BNF with common bean has been considered traditionally low, requiring supplementation with N-fertilizers [5,6]. The failure has been attributed to the high promiscuity of the host plant, interacting with a variety of indigenous rhizobial lineages well established in the soils but with low BNF efficiency [6,7]. Despite reports showing the feasibility of increasing BNF rates in common beans by inoculation with elite *Rhizobium* strains [8,9,10], a recent comparison of the contribution of the 11 most cultivated legumes in the world attributed the lowest contribution to this legume, supplying on average of only 38% of the plant’s total need on N [5].

Literature is rich in studies using classical methods of isolation and characterization of bacteria from nodules of soybean and common bean. For example, in Brazil, where soybean is an exotic plant, by analyzing serological properties, most of the isolates from soybean nodules were identified as a few strains of *Bradyrhizobium* spp. introduced as inoculants [11], and similar results were confirmed by using molecular characterization [12,13]. On the contrary, remarkable diversity has been reported in molecular analyses of common bean nodule isolates [14,15,16,17,18].

Bacterial species rarely function isolated but rather in association with other microbial species [19]. Plants carry microbiomes, which house complex communities of different microorganisms, such as bacteria, fungi, protozoa, archaea, and viruses [20]. There is increasing evidence showing that plant microbiomes have critical roles in protecting plants against pathogens, improving plant growth and production, and promoting adaptive advantages. In particular, studies trying to relate agricultural sustainability with bacterial microbiomes are calling attention [21,22].

Identifying differences in the infection process of the host plants by rhizobia and the interactions occurring inside the nodules can help to elucidate mechanisms of response to inoculation and contribute to improvements in the BNF. However, knowledge about the interactions of bacteria in the rhizosphere and inside the nodules is still scarce. Next Generation Sequencing (NGS) analysis offers an excellent opportunity to understand microbial interactions in a variety of plant-microbe interactions, including legumes-rhizobia symbioses. In the near future, the results obtained may allow the manipulation of plants’ bacterial communities by isolating the “core microbiomes” or the “specific microbiomes,” resulting in a better formulation of inoculants [20,23].

This study aimed to compare the microbiomes of soybean and common bean nodules and roots, including non-inoculated and inoculated plants, under non-sterile soil growth conditions, seeking a better understanding of the interactions occurring in these two contrasting symbioses.

## 2. Results

### 2.1. Relative Abundance of Bradyrhizobium on Soybean Nodules and Roots

A field experiment was performed with soybean in southeast Brazil, and non-sterile soil from southern Brazil was used for the experiment with common beans (Appendix A). Both experiments were performed with sandy soils (Appendix A). Experiments were performed to evaluate the microbiomes of nodules and roots of both soybean and common bean and the effects of inoculation with elite host-specific rhizobial strains and co-inoculation with *Azospirillum brasilense*.

The results obtained in all treatments and replicates are shown in Figure 1 and highlight the greater number and abundance of bacterial genera detected in the roots of both legumes in comparison to the nodules.

From the data obtained, we estimated the relative abundance percentages of *Bradyrhizobium* in the microbiomes of soybean nodules and roots of the treatments: (i) control (non-inoculated); (ii) inoculated with *Bradyrhizobium japonicum* strain SEMIA 5079 and *Bradyrhizobium diazoefficiens* strain SEMIA 5080; (iii) co-inoculated with the two strains of *Bradyrhizobium* and *A. brasilense* strains Ab-V5 and Ab-V6. The combination of strains of *Bradyrhizobium* and *Azospirillum* reflects those used in the great majority of the commercial inoculants used in Brazil. The results are shown in Table 1. The soil where the experiment was performed had been cropped with soybean and inoculated before and showed a compatible rhizobia population estimated at 7.6 × 10^1^ viable cells g^−1^ soil (Appendix A). Although considered very low in comparison to the majority of the soils cropped to soybean in Brazil, e.g., [24], probably because of the stressful conditions of sandy soil (Appendix A), this population was capable of nodulating soybean and resulted in the detection of *Bradyrhizobium* in nodules and roots of non-inoculated plants (Table 1).

The relative abundance of *Bradyrhizobium* sequences was consistently high (>90%) in the nodule microbiome across all treatments (Table 1). In the non-inoculated control, *Bradyrhizobium* represented, on average, 97.62% of the sequences, with a slight increase to 99.20%, not statistically different, when single inoculated with *Bradyrhizobium*. Co-inoculation with *Azospirillum* significantly decreased the relative abundance to 95.88%, compared to the single inoculation with *Bradyrhizobium*.

In the root microbiome, a smaller percentage of *Bradyrhizobium* was observed in the non-inoculated control treatment, on average 7.19%, indicating the presence of other(s) microorganism(s) (Table 1). Inoculation with *Bradyrhizobium* increased the relative abundance of *Bradyrhizobium* to 11.24%, although not statistically significant. Contrarily to the nodule microbiome, the highest relative abundance of *Bradyrhizobium*, of 20.10%, was observed in the co-inoculated treatment, statistically higher than the non-inoculated control.

### 2.2. Microbial Genera Identified in the Microbiomes of Soybean Nodules

A total of 49 different genera were identified in soybean nodules (Figure 2). Their taxonomies are shown in Figure 3, and their percentages are shown in Appendix A. Of the genera identified, 23 were present in the non-inoculated control treatment, 19 when inoculated with *Bradyrhizobium*, and 39 when co-inoculated with *Azospirillum*.

By comparing the genera present in each treatment, nine were detected in all three treatments, eight were present in the nodules of the non-inoculated control and co-inoculated treatments, and six in nodules of the inoculated and co-inoculated treatments (Figure 2). Other genera were unique to each treatment. Among them, six in nodules of the non-inoculated control, four in the treatment inoculated with *Bradyrhizobium*, and 16 in the treatment co-inoculated. It is worth mentioning that some genera that may contribute to important properties and nitrogen fixation are pooled together in the classification, such as the *Burkholderia-Caballeronia-Paraburkholderia* clade, which was present exclusively in the co-inoculated treatment (Figure 3). Other potential nitrogen fixers, such as *Cupriavidus*, *Devosia*, and *Paenibacillus*, were also present exclusively in the co-inoculated treatment.

In some cases, it was not possible to identify the proper taxonomic classification of the bacterial genera. A total of seven phyla, classes, families, and/or orders were identified in the microbiome of soybean nodules. The results, in percentage, are shown in Appendix A. The composition changed with the treatment. For example, UnC Saccharimonadales and UnC Micropepsaceae were present only in the co-inoculated treatment. Unknown diversity in soybean nodule microbiomes was confirmed by the presence of unassigned genera in all treatments (Appendix A).

### 2.3. Microbial Genera Identified in the Microbiomes of Soybean Roots

The number and relative abundance of other bacterial genera detected in soybean root microbiomes were three times higher than in the nodules, as visualized in Figure 1. A total of 145 genera were identified in root microbiomes (Figure 2). In the non-inoculated control encompassed 135 bacterial genera, inoculation with *Bradyrhizobium* resulted in 86, and co-inoculation with *Azospirillum* in 84 genera.

The genera detected in each treatment are shown in Figure 4, and their percentages are shown in Appendix A. Considering the genera occurring in two or three treatments, 70 were present in all three treatments, including several nitrogen fixers, such as *Rhizobium*, *Bradyrhizobium*, *Burkholderia-Paraburkholderia*, and *Cupriavidus*, in addition to other plant-growth promoters as *Pseudomonas*. Another 12 genera were detected in the non-inoculated and inoculated treatments, six in the non-inoculated and co-inoculated treatments, and only two in the inoculated and co-inoculated treatments, *Psychrobacter* and *Brevundimonas*.

Some genera occurred exclusively in one treatment, and the non-inoculated control outstood with 47 unique genera, and among them, we can cite as examples *Rhizobacter*, *Shinella*, *Herbaspirillum*, and *Paenibacillus* (Figure 4). Treatments inoculated with *Bradyrhizobium* had two exclusive genera, *Nevskia* and *Clostridium*, and six when co-inoculated with *Azospirillum*, including *Curtobacterium*, *Sphingobacterium*, *Roseateles*, *Leptothrix*, and *Blastocatella.*

It is worth mentioning that *Azospirillum* was not detected either in the nodule or in soybean root microbiomes.

### 2.4. Relative Abundance of Rhizobium on Common Bean Nodules and Roots

For the common bean, the treatments consisted of (i) control (non-inoculated); (ii) inoculated with *Rhizobium tropici* strain CIAT 899; and (iii) co-inoculated with *R. tropici* and *A. brasilense* strains Ab-V5 and Ab-V6. The *Rhizobium* strain and the combination of *Azospirillum* strains represent those used in the great majority of the commercial inoculants for the crop in Brazil. The results of relative abundance obtained in each replicate may be visualized in Figure 1.

The results indicated the abundant presence of *Rhizobium* in the non-inoculated control treatment in all replicates of the microbiomes of nodules (Figure 1, Table 2). This population should be mainly attributed to indigenous rhizobia, broadly found in Brazilian soils, including the soil used for the experiment, with a population estimated at 4.6 × 10^4^ compatible common bean rhizobia per g of soil (Appendix A).

*Rhizobium* was abundantly found in the microbiomes of common bean nodules, with no statistical difference between the treatments (Table 2). Regarding the roots, as for the soybean (Table 1), the root microbiome of the common bean showed a lower relative abundance of the specific microsymbiont (Table 2, Figure 1), confirming a higher relative abundance of other bacteria than in the nodule microbiome. In the common bean root microbiomes, a lower percentage of *Rhizobium* was observed in the non-inoculated control, on average 13.49%, significantly increasing both in the inoculated (22.41%) and the co-inoculated (24.51%) treatments (Table 2).

### 2.5. Microbial Genera Identified in the Microbiomes of Common Bean Nodules

Considering all treatments, 49 different genera were identified in the microbiomes of common bean nodules (Figure 2), and their percentages are shown in Appendix A. Thirty-five genera were identified in the control, 33 in the inoculated, and 27 in the co-inoculated treatments (Figure 2).

Concerning the genera occurring in more than one treatment, 18 were present in the nodules of all three treatments, including *Rhizobium*, *Bacillus*, *Bradyrhizobium*, and *Devosia*, and the other six were present in both control and inoculated treatments, including *Caulobacter*, *Nordella*, three in the control and co-inoculated, *Hyphomicrobium*, *Pelomons*, *Rhizobacter* and only one in the inoculated and co-inoculated treatments, *Chryseolinea* (Figure 5).

Other genera were specific to the nodule microbiomes of each treatment. Among them, eight were in the non-inoculated control, including *Methylobacterium* and *Flavisolibacter*, eight were in the treatment inoculated with *R. tropici*, *Brevundimonas*, and *Lysobacter*, and five in the co-inoculated with *A. brasilense*, represented by *Massilia* and *Pseudarthrobacter* (Figure 5).

For some bacteria in the microbiome of common bean nodules, it was not possible to reach the taxonomic level of genus, including six phyla, classes, families, and/or orders (Appendix A). Some occurred in more than one treatment, e.g., UnC Gammaproteobacteria I.S. and Unc Oligoflexales were present in the nodules of all three treatments, while others were specific, e.g., Unc Microscillaceae were observed only in the co-inoculated treatment. From 0.06 to 0.86% of the sequences could not be assigned to any known taxonomic classification.

### 2.6. Microbial Genera Identified in the Microbiomes of Common Bean Roots

As observed for the soybean (Table 1), the root microbiome of the common bean harbored a greater number and relative abundance of genera than in the nodules (Table 2, Figure 1). A total of 118 genera were identified in the common bean root microbiomes (Figure 2), and their percentages are shown in Appendix A. In the control treatment, 110 genera were identified, 105 when inoculated with *Rhizobium*, and 105 in the treatment co-inoculated with *Rhizobium* and *Azospirillum* (Figure 2).

Considering the common genera found in more than one treatment, 92 were detected in all three treatments and included several nitrogen fixers, such as *Bradyrhizobium*, *Paraburkholderia*, *Mesorhizobium*, *Cupriavidus*, as well as other plant-growth promoters, such as *Pseudomonas* and *Bacillus* (Figure 6). Another eight genera were detected in the non-inoculated control and inoculated with *R. tropici* treatments, including *Rhodopseudomonas*, *Flavobacterium*, and *Pseudoxanthomonas*, six in the non-inoculated control and co-inoculated with *A. brasilense*, including *Bryobacter*, *Lechevalieria*, and *Lefisonia*, and four in the inoculated and co-inoculated treatments, including *Cellulosimicrobium* and *Xenophilus* (Figure 6).

Some genera were exclusive to one treatment, including four in the non-inoculated control, *Gaiella*, *Lapillicoccus*, *Blastococcus*, and *Mitsuaria*, one in the treatment inoculated with *Rhizobium*, the genus *Chryseobacterium*, and three in the treatment co-inoculated with *Azospirillum*, comprising *Microbispora*, *Archangium*, and *Bauldia* (Figure 6).

Unassigned genera were highly present in all common bean root microbiomes, ranging from 12.50 to 22.97% (Appendix A). A total of 30 phyla, classes, families, and/or orders present in the common bean root microbiomes could not be assigned at the genus level. They were present in all treatments and in different combinations.

As observed in the soybean microbiomes, *Azospirillum* was also not detected in nodules or root microbiomes of the common bean of any treatment.

### 2.7. Diversity of Microbiomes from Soybean and Common-Bean Nodules and Roots

We proceeded with the analyses to improve our understanding of how complex those microbiomes could be and whether they would assemble different communities regarding the treatments. Shannon and Inverted Simpson (InvSimpson) highlighted the lower complexity of nodule microbiomes from both plants, while higher diversities were observed in roots (Table 3, Figure 7). Overall, inoculation treatments reduced diversity measured by both indexes in the roots (Shannon, *p* = 0.002; InvSimpson, *p* = 0.001; Appendix A); however, the reduction was more prominent in common beans than in soybeans (Figure 7B,D). That could indicate that common bean microbiome assemblies in roots are more prompt to change than in soybean, in accordance with the known promiscuity of that legume.

Consistent with the more drastic reduction in alpha diversity with inoculation in common bean roots, beta diversity indicated that microbiomes of common bean roots were more affected by inoculation than those of soybean (Figure 8). The composition of the soybean root microbiome was not significantly affected by the inoculation and remained more diverse, as shown by alpha diversity (Figure 8A). Meanwhile, treatments significantly changed the composition and reduced the alpha diversity of the common bean root´s microbiome (Figure 8C).

Permanova demonstrated significant differences within treatments (*p* = 0.046); however, interactions within inoculation treatments and roots and nodules were not significantly different at a 5% *p*-value threshold (*p* = 0.084; Appendix A). A closer look into the microbiomes from each treatment demonstrated that, in general, beta diversity distances were higher in soybean (Figure 9A) than in common beans (Figure 9B). Still, variations in soybean were much more prominent than in common bean, reinforcing that soybean roots became more diverse than those of common bean (Figure 9). That could indicate that the core root microbiome of the common bean changed less than the soybean microbiome.

## 3. Discussion

For more than a century, nitrogen-fixing nodules were thought to be uniquely inhabited by rhizobia, and culture-based studies have provided important information about the benefits of rhizobia, including differences in the efficiency of the BNF process among legumes and also between strains of the same host legume, e.g., [1,2,3,4,5]. It was only recently that the first genomic studies reported complex microbial communities inside the nodules [25,26,27,28], which was confirmed in our study in nodule microbiomes of both soybean and common bean. However, despite evolving as unique ecological niches for nitrogen fixation through the symbiotic relationship between the host legumes and rhizobia, the biological implications of sharing the nodule environment with other bacteria are not well understood yet. One hypothesis is that other bacteria living inside the nodules and that probably entered the nodule together with rhizobia are endophytes that can help in plant-growth promotion by other microbial processes, such as the synthesis of phytohormones, antimicrobial molecules, siderophores, mineral solubilization capacity, among others [27,28,29,30]. As an example of benefit, an interesting study of nodule endophytes performed with *Lotus burtii* pointed out that in healthy nodules of this legume, *Pseudomonas* species were the prevalent non-rhizobia, and when used as inoculum infected the plant together with a beneficial *Mesorhizobium*, but not with an ineffective *Rhizobium*, benefiting the symbiosis by decreasing the number of ineffective nodules [31].

We investigated two contrasting symbioses, a specific host legume-rhizobia partnership known by a high capacity of BNF, represented by the soybean-*Bradyrhizobium*, e.g., [3,4,5,11,24], and the opposite pair of common bean-*Rhizobium*, broadly known for been promiscuous with several rhizobial species and with low capacity of BNF, e.g., [3,5,6,7,10,14,16,18]. In both cases, we detected that, despite the dominance of the specific rhizobia, nodule microbiomes encompassed several other genera of bacteria. A total of 49 genera were detected in the microbiome of both soybean and common bean nodules, reinforcing the intriguing question about the role of these microbes in this environment that for decades was thought of as being uniquely occupied by the nitrogen-fixing symbionts. In addition, we detected an outstanding number of genera in the root microbiomes, 145 in soybeans and 118 in common beans.

In soybean nodules, the main symbiont *Bradyrhizobium* represented more than 90% of the relative abundance in the microbiome, similar to other reports [32,33], but we were the first to report, with plants grown in non-sterile soil conditions, a slight, although a not statistically significant increase in *Bradyrhizobium* relative abundance in the nodules with single inoculation, from 97.6 to 99.2%, but decreasing to 95.9% with the co-inoculation with *A. brasilense*.

The results of co-inoculation were surprising, as the technology has been applied very successfully in Brazil, with reported increases in nodulation, grain yield, and economic profits for the farmers, and nowadays applied in more than 10 million hectares [34,35,36,37]. We may attribute the decrease observed in our study in the relative abundance of the specific symbiont in nodules with the co-inoculation to competitiveness at the rhizosphere level, as a high concentration of two bacterial species was provided, in comparison to the high concentration of a unique species in the single inoculation. It is also worth mentioning that although endophytes can colonize the apoplast of nodules, while rhizobia infect nodules intracellularly, indicating that they might be compatible in the nodule, they may compete for energy and nutrients. Therefore, to be advantageous, endophytes should provide other benefits to the plants that need to be investigated. However, the results call attention to the need to perform studies with inoculants carrying mixtures of microorganisms, a tendency in the market of biologicals, to avoid competitiveness in nodule microbiome, impairing the benefits of BNF. Effects of inoculation and co-inoculation of soybean on nodule microbiomes were also confirmed by UniFrac analysis si, which detected differences due to the inoculation.

Other nitrogen-fixing genera not reported as symbionts of soybean were also detected in nodule microbiomes, including several rhizobial genera, confirming results reported in previous studies [32,33]. In addition, in our study, other potential diazotrophic bacteria, such as *Methylobacterium*, *Paraburkholderia*, and *Cupriavidus*, were detected on nodule microbiomes, delineating that their contribution to the N nutrition deserves further investigation; these three genera might also be symbiotic to the soybean. One possible role relies on the increasing number of studies reporting other plant-growth-promoting properties in rhizobia, e.g., [31,38,39], indicating that their benefits as endophytes might go far beyond BNF.

We also found as endophytes of soybean nodules several non-rhizobia genera with reported plant-growth promoting capacity, e.g., *Novosphingobium*, *Pseudomonas*, *Bacillus*, *Stenotrophomonas*, and *Sphingomonas*. Other studies reporting non-rhizobia in soybean nodules include *Pseudomonas* [25,40,41,42], *Novosphingobium* [40], *Bacillus* [25,42,43], *Stenotrophomonas* [44], among others [32]. Other non-rhizobia genera were detected in lower relative abundance and usually not in all treatments, including *Enterobacter* and *Escherichia-Shigella*, which might indicate that they are not usual endophytes [32].

In the microbiome of common bean nodules, the relative abundance of the main symbiont *Rhizobium* was higher than 80% and, on average, lower than in soybean. In addition, different from what was observed with soybean, the relative abundance of *Rhizobium* did not differ among the treatments, non-inoculated control, inoculated, or co-inoculated, comprising 91.2, 90.4 and 93.1%, respectively. The lower relative abundance of the main symbiont, in addition to the inability to increase the relative abundance with inoculation, might represent one factor limiting the infection by elite strains and the often-reported lack of response to inoculation [6,7]. In addition, contrarily to the observations on soybean, no differences were observed, in the UniFrac analysis, in common bean nodules as an effect of either inoculation or co-inoculation. The higher relative abundance of other genera in common bean nodules also confirms the high promiscuity of the legume, detected in studies using classical isolation procedures, e.g., [14,18,44,45], and now confirmed by sequencing of the whole nodule microbiome. The common bean microbiome also included several other alpha- and beta-rhizobia, in agreement with reports of isolation of these genera using culture-based methods, e.g., [46,47].

Other non-rhizobia genera with potential plant-growth-promoting properties were also detected in the microbiome of common bean nodules, including some also found in soybean nodules, such as *Novosphingobium* and *Bacillus*, in addition to others reported in nodules of other legumes, such as *Chryseobacterium* in cowpea nodules [48]. Other genera found that might contribute to growth promotion were *Sphingomonas*, *Streptomyces*, and *Devosia*, among others.

The possibility of identifying non-cultivable bacteria using genomic approaches has revealed a great diversity of root microbiome of several plant species, including the *Arabidopsis* model [49] and important grasses such as maize (*Zea mays* L.) [50]. However, the importance of our study is that there are still few studies investigating the roots and nodules of legumes simultaneously.

Some studies indicate that the root microbiome is mainly transferred horizontally, i.e., it derives from the rhizosphere, but it can also be transmitted vertically by seeds [51,52]. Analysis of the microbiomes of soybean and common bean grown in a sterile environment showed that genera such as *Rhizobium*, *Pseudomonas*, *Klebsiella*, *Massilia*, *Acidovorax*, *Stenotrophomonas*, *Methylobacterium*, and *Serratia* can be transferred vertically from the seeds to the root system of these legumes [53]. Indeed, most of these reported genera were also found in the microbiome of soybean and common bean nodules in our study. However, as our study was performed with non-sterile soils, it was not possible to confirm whether they had been acquired by horizontal or vertical transmission.

Several factors are required for introducing bacteria into the plant microbiome from the rhizosphere, including growth rate, competitiveness, chemotaxis, motility, quorum sensing, biofilm formation, adherence, carbon sources, and nutrient rates. In addition, to compose a plant´s microbiome, the bacteria need to be able to adhere and proliferate on the root surface and penetrate cell walls without being tied down by the plant’s immune system [54]. Our study identified a variety of rhizobia and non-rhizobia in the microbiomes of soybean and common bean roots. In previous studies, the analysis of different common bean genotypes identified *Rhizobium*, *Bradyrhizobium*, and *Mesorhizobium*, and non-rhizobia *Novosphingobium*, *Ralstonia*, *Enterobacter*, *Methylophilus*, and *Sphingomonas* genera, and the authors concluded that community structures were not affected by the root genotype [55]. In another study, also several rhizobia and non-rhizobia bacteria were identified in the common bean root microbiome, including *Rhizobium*, *Bradyrhizobium*, *Burkholderia*, *Novosphingobium*, *Arthrobacter*, *Bacillus*, *Caulobacter*, *Pseudomonas*, *Flavobacterium*, *Niastella*, and *Streptomyces* [56]. In soybean, *Bradyrhizobium*, *Rhizobium*, and various non-rhizobia have also been reported, including *Pseudomonas*, *Bacillus*, *Klebsiella*, *Burkholderia*, *Chryseobacterium*, *Variovorax*, *Stenotrophomonas*, *Pantoea*, and *Enterobacter* [57]. The great majority of these genera were also detected in our study. In addition, in another study searching for the core microbiome of the common bean rhizosphere by comparing samples from the USA and Colombia [58], several of the reported genera were also identified in the root microbiomes of our study, and both studies confirm a large number of under-described or unknown members.

Our study has shown a higher relative abundance of genera in roots than in nodules, also confirmed in the alpha- and beta-diversity analyses. We should also comment that in a study with common bean [54], the microbiome of the rhizosphere was greater than in the endosphere, composed of seeds and roots, altogether indicating that diversity decreases to more specific communities from the rhizosphere to the nodule.

Noticeable is that inoculation and co-inoculation of both soybean and common bean had greater effects on the relative abundances of *Brayrhizobium* and *Rhizobium* in soybean and common bean root microbiomes, respectively, that in nodule microbiomes (Table 1 and Table 3). Also, inoculation reduced the number of unique genera in the root microbiomes in comparison to the non-inoculated control, with an emphasis on soybeans. This result highlights that inoculants may cause shifts in the natural microbial communities and in root microbiomes, calling attention to the importance of using only elite strains with proven low impact in other microbial processes that may be critical for plant growth. For example, there are results showing that inoculation of *Bacillus amyloliquefaciens* or *Bacillus subtilis* affected the bacterial diversity in the rhizosphere microbiome of wheat (*Triticum aestivum* L.) [59], tomato (*Solanum lycopersicum* L.) [60], and cucumber (*Cucumis sativus* L.) [61]. In another report, co-inoculation of *Bradyrhizobium japonicum* 5038 and *Bacillus aryabhattai* MB35-5 affected rhizospheric diversity in soils of China, suggesting the feasibility of rebuilding the microbial community via inoculation with specific strains [62]. In the analyses of both alpha- and beta-diversities, decreases in root microbiome diversity were shown in response to inoculation, being more prominent in common beans than in soybeans.

The analysis of the maize rhizospheric community of plants inoculated with *Azospirillum argentinense* strain Az39 indicated that the colonization with the bacterium induced an increase in the relative abundance of beneficial bacterial genera [63]. Also, inoculation of *Azospirillum lipoferum* strain CRT1 in maize seeds resulted in a significant change in the composition of the native rhizobacteria community [64]. Very interesting, *Azospirillum* was not detected in any of the microbiomes investigated in our study, including the co-inoculation treatments with this genus. This result reinforces that *A. brasilense* strains Ab-V5 and Ab-V6 perform their main roles at the rhizosphere level, not as endophytes. A study performed with these two strains confirmed that they were unable to colonize internally leaves [65], indicating that they may be very specific for root rhizosphere. The inoculation of *A. argentinense* strain Az39 also indicated that the genus became the most abundant in the maize rhizosphere [63].

The root and nodule microbiomes impact plant growth because they are related to the function of the associated microbial communities. Detailed knowledge of these microbial communities is, therefore, necessary to exploit their benefits for plant growth [66]. Biocontrol activities against invading pathogens and diseases through the production of antibiotics, lytic enzymes, volatile pathogen-inhibiting compounds, and siderophores are other important benefits to the plant. Some bacteria protect the plant from pathogens by modulating the level of plant hormones and inducing systemic plant resistance; in particular, genera such as *Pseudomonas*, *Streptomyces*, *Bacillus*, *Paenibacillus*, *Enterobacter*, *Burkholderia*/*Paraburkholderia* have been reported for their role in suppressing pathogens [51]. By suppressing pathogens, nitrogen fixation by the specific rhizobia may be highly favored, bringing more benefits to plant growth. Consequently, it is important to understand the role of these microorganisms in growth promotion and disease control and their application as biofertilizers and biopesticides [67]. We also found several nitrogen fixers and phytohormone producers in nodule and root microbiomes of both legumes and, similar to the core rhizosphere microbiome of common bean described by Stopnisek and Shate [58]. Plants benefit from many microbial processes performed in their microbiomes, explaining why they invest part of their carbon and other nutrient sources in microbial maintenance [20]. For example, the endophytic consortium of *Bacillus* sp. and *Pseudomonas* sp. has been shown to significantly increase the phosphate solubilization efficiency of wheat cultivars that were growing in phosphate-deficient soil [68]. In addition, the tolerance of some plant species to drought was positively correlated with the increase in the relative abundance of the genus *Streptomyces* [69]. Despite knowing the plant growth-promoting ability of these genera, understanding their benefits on the microbiome of soybean and common bean roots may maximize their benefits to agriculture.

Our main goal in this study was to add information that could contribute to explaining the differences between the very specific and efficient soybean-*Bradyrhizobium* and the less specific and efficient common bean-*Rhizobium* symbioses. Important results were obtained, including that in the more efficient symbiosis, the specific genus *Bradyrhizobium* had greater relative abundance in the nodule microbiome, with the possibility of increasing by inoculation of elite strains. In common bean, the main symbiont *Rhizobium* was not only in lower relative abundance in nodule microbiomes but also no increases were observed with inoculation. Noticeably, the inoculation of both soybean and common bean plants resulted in significant increases in the relative abundance of *Bradyrhizobium* and *Rhizobium* in root microbiomes, respectively, and co-inoculation with *A. brasilense* highly contributed to this increase. If this represents a strategy to get closer to occupying nodule microbiomes at any opportunity, it deserves further investigation. Our results also highlight the importance of investigating the impact of inoculation on root microbiomes, as both alpha- and beta-diversity were reduced by inoculation, being more prominent in common beans, which may affect the contribution of other microbial processes. Maybe our study has raised more questions than it has answered, but certainly, it contributed to important information clarifying differences in symbiotic performances.

## 4. Material and Methods

### 4.1. Plant Growth Conditions

#### 4.1.1. Soybean

One field experiment was performed with soybean at the farm “Bela Vista” in Lutécia, state of São Paulo, southeast Brazil. Information about the localization, geographic coordinates, altitude, and climate is shown in Appendix A. Forty days before sowing, 20 soil subsamples were taken at the 0–20 cm and 20–40 cm depth layers for chemical and granulometric analyses. Analyses were performed as described before [24], and the results are displayed in Appendix A. The soybean-nodulating rhizobial population was assessed only at the 0–10 cm topsoil layer in the same samples used for chemical analysis. The quantitative population was estimated by the most probable number (MPN) method with plant counting [70], using soybean cv. BRS 1010 IPRO to trap rhizobia and is also shown in Appendix A. Chemical correction of the soil before sowing was performed as described before [24]. Immediately before sowing, fertilization with 300 kg ha^−1^ of the formulation 00-20-20 (0 N, 60 kg ha^−1^ of P_2_O_5_ and 60 kg ha^−1^ of K_2_O) was applied in-furrow in all treatments.

All strains used in the experiments are deposited at the “Diazotrophic and Plant Growth Promoting Bacteria Culture Collection of Embrapa Soja” (WFCC Collection # 1213, WDCM Collection # 1054), in Londrina, State of Paraná, Brazil.

For the soybean experiment, 2 inoculants were prepared. The first included *B. japonicum* strain SEMIA 5079 (CPAC 15 and CNPSo 06) and *B. diazoefficiens* strain SEMIA 5080 (CPAC 7 and CNPSo 07). Strains were grown separately on a modified-YM medium [71] till the end of the exponential growth phase. The concentration of each strain was adjusted to 2 × 10^9^ CFU (colony forming units, representing the number of viable cells) mL^−1^, and they were mixed and applied to supply 1.2 million cells seed^−1^, as recommended for the crop in Brazil [24]. It is worth mentioning that commercial inoculants for the soybean in Brazil always carry 2 *Bradyrhizobium* strains, and the combination used in our experiment represented more than 90% of the 80 million doses sold in the last crop season [4,11,24]. For the inoculant with *A. brasilense* strains Ab-V5 (CNPSo 2083) and Ab-V6 (CNPSo 2084), they were grown separately on DYGS medium [72] till the end of the exponential growth phase, concentrations were adjusted to 2 × 10^8^ CFU mL^−1^ and applied to supply 120,000 cells seed^−1^, as recommended for the crop in Brazil. This combination of strains was used in almost 100% of the 10 million doses commercialized in the last crop season in Brazil [34,35,37]

The experiment was composed of 3 treatments, (i) non-inoculated control, (ii) inoculated with *Bradyrhizobium*, and (iii) co-inoculated with *Bradyrhizobium* and *A. brasilense*. Inoculation was performed on seeds of cv. BRS 1010 IPRO immediately before sowing. The experiment was performed in a randomized complete block design (RCBD) with 6 replicates. Plot sizes, conduction of the experiment, and other procedures were as described before [24].

At 40 days after emergence, 5 plants were randomly harvested from each plot for the analyses. They were pooled in 1 sample per replicate of each treatment.

#### 4.1.2. Common Bean

The experiment with common beans was performed in pots filled with non-sterile soil, and plants were grown under greenhouse conditions. In the past 2 years, we had many pest and disease problems with the common bean crop in Paraná State, southern Brazil, demanding high application of fungicides and insecticides, negatively impacting the symbiosis. Therefore, we preferred to collect the soil and perform the study under greenhouse conditions, avoiding pests and diseases. Soil for the experiment was collected in Ponta Grossa, Paraná State. Localization, geographic coordinates, altitude, and climate are shown in Appendix A. Soil chemical and granulometric properties are shown in Appendix A. The common bean nodulating rhizobial population was assessed by the MPN [70] method with plant counting on cv. Carioca. For the experiment, pots were filled with 5 kg of soil after correction, as described by [24]. Before being used to fill the plots, the soil received 300 kg of N-P-K (60 kg ha^−1^ of P_2_O_5_ and 60 kg ha^−1^ of K_2_O).

Two inoculants were also prepared for the common bean experiment. The first was with *R. tropici* CIAT 899 (SEMIA 4077), grown in a modified-YM medium [71] until the late exponential stage, and the concentration was adjusted to 2 × 10^9^ CFU mL^−1^ and applied to supply 1.2 million cells seed^−1^. In Brazil, commercial inoculants for the common bean carry only 1 strain that must belong to the *R. tropici* group, and CIAT 899 is prevalent in commercial inoculants [8,9,10]. The inoculant of *A. brasilense* strains CNPSo Ab-V5 + Ab-V6 was prepared as described in the soybean experiment.

The experiment was performed with cv. Carioca and composed of 3 treatments: (i) non-inoculated control; (ii) inoculated with *Rhizobium*; and (iii) co-inoculated with *Rhizobium* and *A. brasilense*.

The experiment was performed in a randomized complete block design (RCBD) with 6 replicates. The growth conditions at the greenhouse experiment were as described before [73].

At 35 days after emergence, 5 plants were randomly harvested from each replicate for the analyses. They were pooled in 1 sample per replicate of each treatment.

### 4.2. DNA Extraction, Sequencing, and Microbiome Analysis

#### 4.2.1. Samples Preparation

In the laboratory, the soybean and common bean plants were surface-sterilized with 70% alcohol and 5% hypochlorite, followed by washing with sterile distilled water, as described by [71]. Plant tissues were divided into nodules, roots, and shoots and kept at −80 °C. Simultaneously, to confirm the effectiveness of the surface-sterilization process, dozens of randomly chosen nodules and roots were added to plates containing modified-YM-agar, DYGS-agar, and potato-dextrose-agar media to verify the growth of any contaminants. We confirmed the effectiveness of the sterilization process. Frozen roots and nodules were macerated using liquid nitrogen and kept again at −80 °C after maceration.

#### 4.2.2. DNA Extraction

To extract DNA from roots and nodules of soybean and common bean, the DNeasy Plant kit from QIAGEN was used. The macerated samples were weighed, and 100 mg of each sample was transferred to an Eppendorf tube, and extraction followed the manufacturer’s instructions.

#### 4.2.3. Preparation of 16S rRNA Amplicon Libraries and Sequencing

The libraries were submitted to 2 rounds of PCR reactions. In the first, the reaction was performed to amplify the V3-V4 region of the 16S rRNA gene using primers 341F CCTACGGGNGGCWGCAG-30 and 785R GACTACHVGGGTATCTAATCC [74] and chloroplast and mitochondria amplification blocking primers, PNA clamps [75]. The reaction consisted of: 2.5 µL of DNA; 12.5 µL of 2× Kappa Hifi HotStart Mix; 0.2 µM of each primer, and 0.25 µM of each PNA clamp; with a final volume of 25 µL. The amplification conditions used were: 95 °C for 3 min; 25 × 95 °C for 30 s, 55 °C for 30 s, 72 °C for 30 s; 72 °C for 5 min. In the second step, the indexes were added to each sample. The reaction consisted of 25 µL of 2× Kappa Hifi HotStart Mix; 5 µL of each index (Illumina TruSeq i5 and i7), and 5 µL of the previously amplified product; final volume of 50 µL. The conditions used were: 1 × 95 °C 3 min; 8 × 95 °C 30 s, 55 °C 30 s, 72 °C 30 s; 72 °C 5 min.

After the PCR reactions, the DNA samples were purified using magnetic beads (Agencourt AMPure XP) on a magnetic rack. They were adjusted to equimolar concentrations that were quantified via qPCR (Collibry Library Quantification Kit). Sequencing was performed on a MiSeq V2-300 platform.

#### 4.2.4. Bioinformatics and Statistical Analysis

The multiplexed raw single-end reads were checked for quality, and the denoise was performed by the DADA2 tool [76] with max_ee = 2 to remove poor-quality reads, singletons, and PCR chimeras. The ASVs and the feature table were then obtained. Only ASVs greater than 270 bases were retained for further analysis. The signature of taxonomies was performed with global alignment using the Vsearch tool [77] against the Silva v132 database, and only taxonomies with ≥99% identity with some sequence in the database were signed. Sequences with ≤99% id were classified as unsigned. Microbiomes and statistics were analyzed with the phyloseq [78], vegan [79], and agricolae [80] R packages. Relative abundance bar plots were created in R using the ggplot2 package. Upsetplots were built by using the ComplexUpset R package [81,82].

Statistical analysis of the relative abundance of *Bradyrhizobium* in the soybean microbiome and *Rhizobium* in the common bean microbiome was tested for normality by the Shapiro-Wilk test and homogeneity by the Bartlett test. The relative abundance percentages of *Bradyrhizobium* and *Rhizobium* were transformed to arcsine (x/100)0.5 before analysis. Means were analyzed by 1-way ANOVA and Tukey’s test (α = 5%). All analyzes were performed in R 4.1.0 (R Foundation for Statistical Computing, Vienna, Austria).

## Figures and Tables

**Figure 1 ijms-23-12035-f001:**
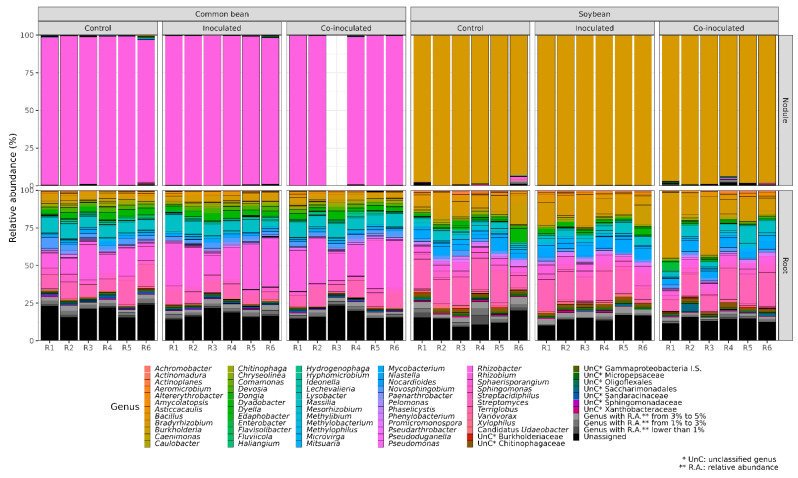
Relative abundance (%) of the genera detected in the microbiomes of soybean and common bean nodules and roots. Treatments included (i) non-inoculated controls; (ii) soybeans inoculated with *Bradyrhizobium japonicum* strain SEMIA 5079 + *Bradyrhizobium diazoefficiens* SEMIA 5080, and common beans inoculated with *Rhizobium tropici* CIAT 899; and (iii) both legumes co-inoculated with the specific rhizobia and *Azospirillum brasilense* strains Ab-V5 and Ab-V6. R1 to R6 refer to six biological replicates of each treatment, and the white column in one nodule of common bean was due to a sample loss.

**Figure 2 ijms-23-12035-f002:**
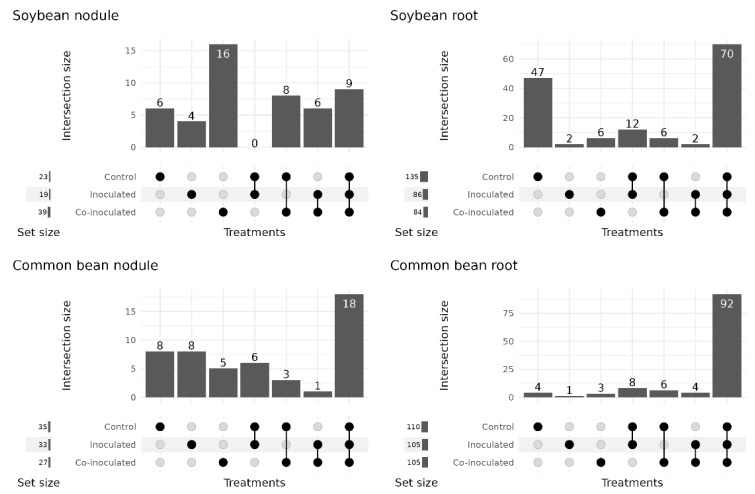
The number of genera found in the microbiomes of soybean and common bean nodules and roots in the treatments (i) non-inoculated (control); (ii) inoculated with the specific rhizobia; (iii) co-inoculated with the specific rhizobia and *Azospirillum brasilense*.

**Figure 3 ijms-23-12035-f003:**
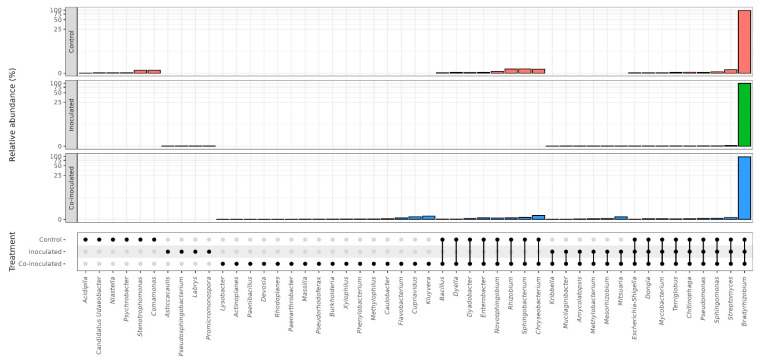
Genera of bacteria in the microbiomes of soybean nodules in the treatments (i) non-inoculated (control); (ii) inoculated with *Bradyrhizobium* spp.; and (iii) co-inoculated with *Bradyrhizobium* spp. and *Azospirillum brasilense*.

**Figure 4 ijms-23-12035-f004:**
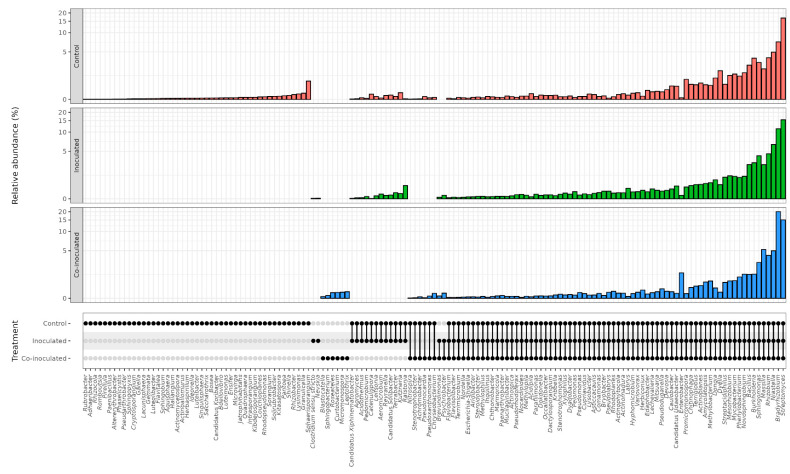
Genera of bacteria in the microbiomes of soybean roots in the treatments (i) non-inoculated (control); (ii) inoculated with *Bradyrhizobium* spp.; and (iii) co-inoculated with *Bradyrhizobium* spp. and *Azospirillum brasilense*. A high percentage of unsigned genera was detected in all treatments, ranging from 8.82 to 19.33% (Appendix A). Considering all identified sequences, 189 distinct taxonomies of bacteria and unassigned genera were observed in the control treatment, 106 in the inoculated treatment, and 105 in the co-inoculated treatment.

**Figure 5 ijms-23-12035-f005:**
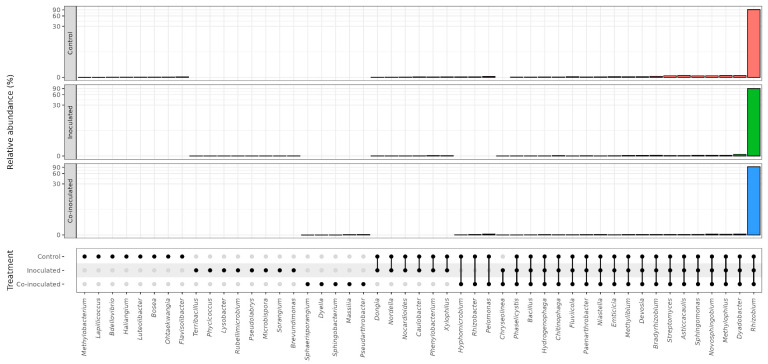
Genera of bacteria in the microbiomes of common bean nodules in the treatments (i) non-inoculated (control); (ii) inoculated with *Rhizobium tropici*; and (iii) co-inoculated with *R. tropici* and *Azospirillum brasilense*.

**Figure 6 ijms-23-12035-f006:**
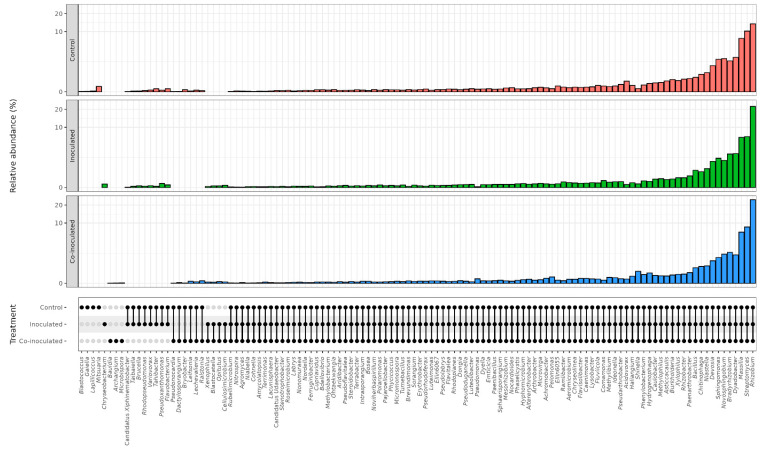
Genera of bacteria in the microbiomes of common bean roots in the treatments (i) non-inoculated (control); (ii) inoculated with *Rhizobium tropici*; and (iii) co-inoculated with *R. tropici* and *Azospirillum brasilense*.

**Figure 7 ijms-23-12035-f007:**
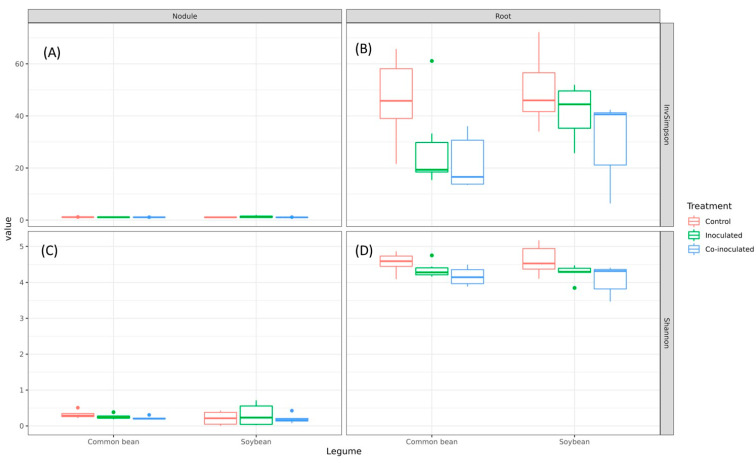
Bacterial Inverted Simpson indexes in the microbiomes of common bean and soybean nodules (**A**) and roots (**B**) and Shannon indexes in nodules (**C**) and roots (**D**) of plants inoculated with specific rhizobia and co-inoculated with rhizobia and *Azospirillum brasilense*.

**Figure 8 ijms-23-12035-f008:**
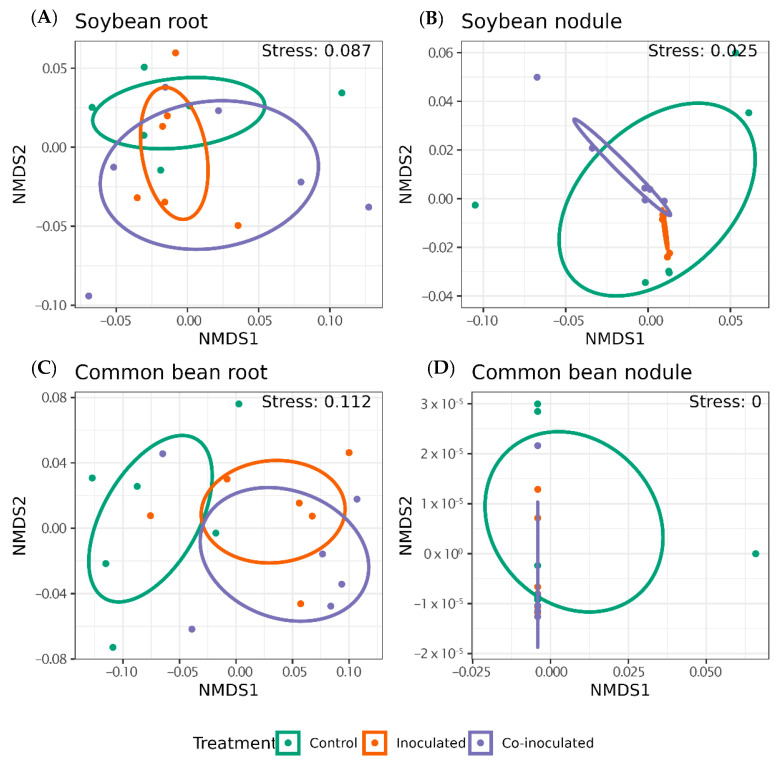
Bacterial weighted UniFrac beta diversity distances in the microbiomes of roots and nodules of common bean and soybean non-inoculated (control). Inoculated with specific rhizobia, and co-inoculated with rhizobia and *Azospirillum brasilense*.

**Figure 9 ijms-23-12035-f009:**
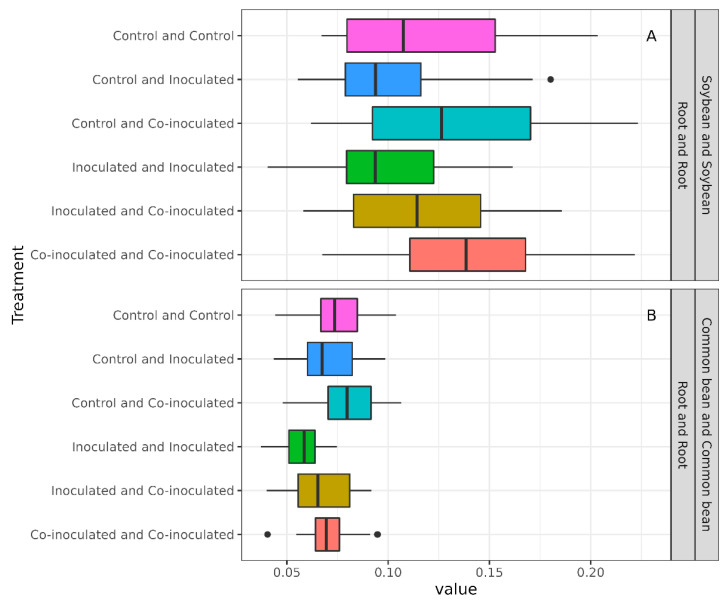
Comparisons between weighted UniFrac distances in the microbiomes of roots and nodules of (**A**) soybean and (**B**) common bean non-inoculated (control), inoculated with specific rhizobia, and co-inoculated with rhizobia and *Azospirillum brasilense*.

**Table 1 ijms-23-12035-t001:** Relative abundance (%) of *Bradyrhizobium* in the microbiomes of nodules and roots of soybean (i) non-inoculated (control); (ii) inoculated with *Bradyrhizobium* spp.; and (iii) co-inoculated with *Bradyrhizobium* spp. and *Azospirillum brasilense*.

Replicate	Nodules			Roots		
	Control ^1^	Inoculated	Co-Inoculated	Control	Inoculated	Co-Inoculated
1	97.76	99.30	96.33	4.59	13.49	41.00
2	99.62	99.16	98.18	10.51	11.31	11.37
3	98.36	99.77	97.55	8.64	18.14	37.55
4	98.52	99.09	92.25	4.41	10.08	9.92
5	99.93	99.58	94.55	9.40	8.50	12.08
6	91.55	98.33	96.45	5.61	5.89	8.66
Mean ^2^	97.62 ^ab^	99.20 ^a^	95.88 ^b^	7.19 ^b^	11.24 ^ab^	20.10 ^a^
SD ^3^	3.08	0.50	2.17	2.65	4.25	14.94

^1^ Soybean-compatible *Bradyrhizobium* population estimated at 7.6 × 10^1^ viable cells g^−1^ soil. ^2^ Means followed by the same letter for the nodule or root microbiomes are not statistically different (*p* < 0.05, Tukey). ^3^ Standard deviation.

**Table 2 ijms-23-12035-t002:** Relative abundance (%) of *Rhizobium* present in the microbiome of nodules and roots of (i) common bean non-inoculated (control); (ii) inoculated with *Rhizobium tropici*; and (iii) co-inoculated with *R. tropici* and *Azospirillum brasilense*.

Replicate	Nodules			Roots		
	Control ^1^	Inoculated	Co-Inoculated	Control	Inoculated	Co-Inoculated
1	90.51	92.95	94.99	9.65	25.67	24.29
2	95.57	89.69	95.79	11.23	24.29	26.02
3	89.67	89.29	- ^2^	21.81	13.08	16.23
4	94.18	91.03	88.46	12.04	18.98	18.48
5	95.83	92.68	89.64	16.77	26.70	31.37
6	81.51	86.75	96.51	9.42	25.76	30.69
Mean ^3^	91.21 ^a^	90.40 ^a^	93.08 ^a^	13.49 ^b^	22.41 ^a^	24.51 ^a^
SD ^4^	5.41	2.33	38.15	4.87	5.34	6.20

^1^ Common bean compatible *Rhizobium* population estimated at 4.6 × 10^4^ viable cells g^−1^ soil. ^2^ Sample lost. ^3^ Means followed by the same letter for the nodule or root microbiomes are not statistically different (*p* < 0.05, Tukey). ^4^ Standard deviation.

**Table 3 ijms-23-12035-t003:** Bacterial alpha-diversity indexes in the microbiomes of roots and nodules of common bean and soybean non-inoculated (control); inoculated with specific rhizobia; and co-inoculated with rhizobia and *Azospirillum brasilense*.

Plant	Tissue	Treatment	InvSimpson		Shannon	
			Mean	SD	Mean	SD
Soybean	Nodule	Control	1.09	0.08	0.21	0.17
		Inoculated	1.31	0.37	0.31	0.28
		Co-inoculated	1.06	0.04	0.24	0.11
	Root	Control	49.86	12.48	4.62	0.38
		Inoculated	41.30	9.47	4.27	0.20
		Co-inoculated	30.71	14.53	4.09	0.38
Common bean	Nodule	Control	1.13	0.03	0.33	0.10
		Inoculated	1.10	0.02	0.26	0.07
		Co-inoculated	1.09	0.01	0.23	0.04
	Root	Control	46.79	15.76	4.56	0.26
		Inoculated	27.92	15.99	4.36	0.20
		Co-inoculated	21.66	9.63	4.18	0.23

## Data Availability

Data are included in the manuscript or will be available upon request.

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
