# Peer review of "Microbiome of Nodules and Roots of Soybean and Common Bean: Searching for Differences Associated with Contrasting Performances in Symbiotic Nitrogen Fixation"

_ijms, 2022, doi:10.3390/ijms231912035_

Round 1
Reviewer 1 Report
This paper looks at the effect of inoculation on the microbial diversity in soybean and common bean nodules and roots (endophytic population). Treatments were no inoculation, inoculation with compatible rhizobia (two Bradyrhizobium strains for soybean; Rhizobium tropici for bean), and inoculation with rhizobia and Azospirillum. Experiments were done in the field for soybean, and in the lab (with soil field samples) for bean. Identity of the microflora was determined by amplification of part of the 16sRNA gene and high throughput sequencing. The results indicated that bradyrhizobia made up the huge majority of the strains in soybean nodules, but there were also apparent endophytes of other genera in these nodules. The community from surface sterilized soybean roots was more complex with lower percentages of bradyrhizobia, and there was a significant effect of inoculation with Azospirillum in addition to the rhizobia. For bean, the occupancy by rhizobia was lower, but the general story much the same. Interestingly Azospirillum itself was not detected in any of the treatments, suggesting it cannot colonize the inside of the root or co-occupy a nodule.
There is useful information here, but I am not sure that the study, other than describing the diversity in these systems, does all that much to address the differences in response to inoculation between bean and soybean.
In addition, I wonder what controls were done to assure that both nodules and roots were completely surface sterilized before extracting DNA, and wonder if some of the diversity being detected might not be due to bacteria that survived the surface sterilization.
Finally, it is not clear to me what the mechanism would be whereby Azospirillum inoculation would alter the composition of the nodule microflora - though the percent changes are small, they appear to be significantly different.
Also there are a number of typos and small grammar and English usage mistakes that should be corrected in any revision. eg. environmef in line 312, also the line 26 should be which not what, line 30 should be "whose" contribution, not which and so on.
Author Response
This paper looks at the effect of inoculation on the microbial diversity in soybean and common bean nodules and roots (endophytic population). Treatments were no inoculation, inoculation with compatible rhizobia (two Bradyrhizobium strains for soybean; Rhizobium tropici for bean), and inoculation with rhizobia and Azospirillum. Experiments were done in the field for soybean, and in the lab (with soil field samples) for bean. Identity of the microflora was determined by amplification of part of the 16sRNA gene and high throughput sequencing. The results indicated that bradyrhizobia made up the huge majority of the strains in soybean nodules, but there were also apparent endophytes of other genera in these nodules. The community from surface sterilized soybean roots was more complex with lower percentages of bradyrhizobia, and there was a significant effect of inoculation with Azospirillum in addition to the rhizobia. For bean, the occupancy by rhizobia was lower, but the general story much the same. Interestingly Azospirillum itself was not detected in any of the treatments, suggesting it cannot colonize the inside of the root or co-occupy a nodule.
There is useful information here, but I am not sure that the study, other than describing the diversity in these systems, does all that much to address the differences in response to inoculation between bean and soybean.
Reply: It is still a paradox that we have two contrasting symbioses as those with the common bean and the soybean. And common bean almost always does not respond to inoculation under non-sterile conditions. We were the first to report, under non-sterile conditions, that inoculation with Bradyrhizobium increased their abundance in nodule microbiome, while in common bean the inoculation did not increase the Rhizobium abundance in the nodule microbiome. This result can be highly related to the lack of response to inoculation of common bean, contrarily to the soybean. We tried to give more emphasis on that. We also believe that it was very important to find such diversity of bacteria inside the nodules. For those such as myself who studied biological nitrogen fixation and symbioses for decades, to see the evolution in our knowledge, once the general belief was that nodules were inhabited exclusively by the specific symbionts, is very important. We discuss our hypothesis, that other bacteria living inside the nodules are endophytes that can help in plant-growth promotion by other microbial processes, such as the synthesis of phytohormones, antimicrobial molecules, siderophores, mineral solubilization capacity, among others. We gave more emphasis to this in the discussion.
In addition, I wonder what controls were done to assure that both nodules and roots were completely surface sterilized before extracting DNA, and wonder if some of the diversity being detected might not be due to bacteria that survived the surface sterilization.
Reply: We added more information to the Material and Methods section. “… to confirm the effectiveness of the surface-sterilization process, dozens of randomly chosen nodules and roots were added to plates containing modified-YMA, DYGS with agar and potato-dextrose-agar medium, to verify growth of any contaminants. We confirmed the effectiveness of the sterilization process
Finally, it is not clear to me what the mechanism would be whereby Azospirillum inoculation would alter the composition of the nodule microflora - though the percent changes are small, they appear to be significantly different.
Reply: We discussed the question. The best explanation would be that with co-inoculation we had two species, the specific rhizobia and A. brasilense, both applied at high concentrations on the seeds. Therefore, there was some degree of competitiveness, resulting in decrease of rhizobia being able to occupy the nodule, in comparison with single rhizobia inoculation. We added this hypothesis to the abstract and to the discussion
Also there are a number of typos and small grammar and English usage mistakes that should be corrected in any revision. eg. environmef in line 312, also the line 26 should be which not what, line 30 should be "whose" contribution, not which and so on.
Reply: All mistakes pointed out were corrected. In addition, we reviewed the English of the whole text.
Reviewer 2 Report
The manuscript by Bender et al. descripted the difference between the soybean-Bradyrhizobium and common bean-Rhizobium symbionts through next generation sequencing (NGS) of nodules and root microbiomes. To well understand the microbial interactions occurring in field, three treatments were investigated, including non-inoculated, inoculated rhizobia and co-inoculated. Some results are interesting, but some place need be improved. In general, the text would merit a very careful reading and substantial corrections.
Major revision
-Fig 1. Need be carefully re-organized. Color of each genus is too closed, it is hard to differentiate from each other.
-There has no more analysis about bioinformatic data, such as α, β diversity or PCoA /NMDS.
-There has no physiological analysis about BNF, Shoot biomass of each treatment.
-There has no compatible relationship between soybean and common bean, for the authors inoculated different rhizobium on these two plants. I suggest the authors may inoculate one rhizobium which can form nodules both on soybean and common bean, such as NGR234.
-I suggest the authors use Venn to display the same and different genera of table 2-5.
-Why the authors inoculated two Bradyrhizobium and two A. brasilense?
Minor revision
-The genus name of each bacterial strains needs in italic.
-Some abbreviation need give full name. Such as in table S2, line 549 PNA, Line 565 ASV.
-Line 558, “PRC”? is “PCR”
-Line 531 At 35 days after emergence, it is better “after inoculation”
-Line 523. A. brasilense strains CNPSo 2083 + 2084 is the same with line 109 A. brasilense strains Ab-V5 and Ab-V6? The same strain need be uniformed.
Author Response
The manuscript by Bender et al. descripted the difference between the soybean-Bradyrhizobium and common bean-Rhizobium symbionts through next generation sequencing (NGS) of nodules and root microbiomes. To well understand the microbial interactions occurring in field, three treatments were investigated, including non-inoculated, inoculated rhizobia and co-inoculated. Some results are interesting, but some place need be improved. In general, the text would merit a very careful reading and substantial corrections.
Major revision
-Fig 1. Need be carefully re-organized. Color of each genus is too closed, it is hard to differentiate from each other.
Reply: We also changed the layout, so that it is now easier to differentiate.
-There has no more analysis about bioinformatic data, such as α, β diversity or PCoA /NMDS.
Reply: We explained and changed throughout the manuscript to clarify that we were not performing a study about diversity in the strictu senso. Our main interest was to identify rhizobia and other genera in nodule and root microbiomes of the two contrasting symbioses, and to verify if there were shifts with inoculation. We believe that it is now clearer.
-There has no physiological analysis about BNF, Shoot biomass of each treatment.
Reply: Our study was performed based on the results reported in dozens of experiments performed for decades. The differences between the two symbioses are well known. Including two important recent reviews, cited in the manuscript (Peoples et al., 2021; Herridge et al., 2022). Therefore, we wanted to focus on the microbiomes. In the discussion, when we mention the differences, we cite five papers for the soybean and eight for the common bean.
-There has no compatible relationship between soybean and common bean, for the authors inoculated different rhizobium on these two plants. I suggest the authors may inoculate one rhizobium which can form nodules both on soybean and common bean, such as NGR234.
Reply: NGR234 is not a good strain neither for common bean, nor for soybean. It nodulates the legumes, but it is not very efficient. We used the best elite strains, with known superior performance worldwide. Rhizobium tropici CIAT 899 is used in commercial inoculants for common beans at least in South America and Africa. And the two Bradyrhizobium strains are used in more than 100 million doses in Brazil, Bolivia, Paraguay.
-I suggest the authors use Venn to display the same and different genera of table 2-5.
Reply: We replace all tables stating genera name by figures. We do believe that it is more appealing. Thanks for the suggestion. We had to re-analyze everything to build the figures, and therefore the taxonomy was updated.
-Why the authors inoculated two Bradyrhizobium and two A. brasilense?
Reply: Because that´s how they are used in the commercial inoculants in Brazil. We included this explanation in the results and in the material and methods section.
Minor revision
-The genus name of each bacterial strains needs in italic.
Reply: Sorry, it was a problem in the transfer of the manuscript to the journal file.
-Some abbreviation need give full name. Such as in table S2, line 549 PNA, Line 565 ASV.
Reply: Corrected. In line 549 the full name was three lines above.
-Line 558, “PRC”? is “PCR”
Reply: Sorry. Corrected.
-Line 531 At 35 days after emergence, it is better “after inoculation”
Reply: It is generally given as days after emergence (DAE) not after inoculation
-Line 523. A. brasilense strains CNPSo 2083 + 2084 is the same with line 109 A. brasilense strains Ab-V5 and Ab-V6? The same strain need be uniformed.
Reply: Sorry, we changed to leave everything as Ab-V5 and Ab-V6
Reviewer 3 Report
In the manuscript entitled “Microbiome of Nodules and Roots of Soybean and Common Bean: Searching for Differences Associated with Contrasting Responses to Inoculation and Efficiency of Biological Nitrogen Fixation” the authors compare the microbiomes of soybean and common bean roots and nodules after different treatments (no inoculation, inoculation with rhizobia, co-inoculation with rhizobia and Azospirillum). Soybean and common bean are both important crops, but they differ in their promiscuity and their capacity to fix nitrogen symbiotically and thus their requirements for additional fertilisation. The authors report that in general the bacterial diversity in roots is higher than in nodules and that common bean nodules have a higher diversity than soybean nodules. They also report that the relative abundance of Bradyrhizobia changes upon co-inoculation with Azospirillum. This work adds to our understanding about how inoculation treatments impact microbial communities. In particular, how inoculation with elite rhizobia and Azospirillum, a widespread agricultural practice in Brazil one of the major soybean producers in the world, could impact root microbiomes.
Overall the design of the sampling is sound, and the description of the methods is detailed. However, the data generated is only superficially analysed and some conclusions are not supported by the data. Writing and data display should be improved. Examples illustrating these points are listed below.
General comments:
1. The motivation for this work is not clearly defined. The authors try to address two loosely related questions:
A) How the nodule microbiome affects symbiotic performance?
B) What is the effect of different inoculation treatments on the root and nodule microbiomes?
Although Question A is in my opinion an extremely important question in the field of BNF, it cannot be answered using the experimental design presented here. Soybean and common bean are two distinct crops that were grown under different conditions (field vs. Greenhouse). Conclusions about the relationship between the microbiome and the symbiotic performance cannot be made.That is why in my opinion the authors should focus on question B. I recommend to re-write the paper to make this research question and its novelty clearer.
2. The writing of the manuscript is not clear. Sentences are often too long and difficult to understand. For more specific examples see specific comments (Point 3).
3. Almost a third (25 out of 77) of the citations used in the manuscript and more that half (13 out of 23) of the citations used in the introduction are authored by the senior author. Important papers in the field are left out. For example, in the introduction (Lines 83-85) the authors state “…knowledge about the interactions of bacteria in the rhizosphere and inside the nodules is still scarce”. However there is a growing number of papers describing the root and nodule microbiomes, even some investigate the interaction between symbiotic performance and the nodule microbiome and the role of non-rhizobial endophytes. See more examples in point 5.
i. Soybean
Cui et al (https://journals.asm.org/doi/epub/10.1128/mSystems.01299-20)
Xia et al (https://onlinelibrary.wiley.com/doi/10.1111/mec.14027)
ii. Bean
Medina-Paz et al (https://www.ncbi.nlm.nih.gov/pmc/articles/PMC9269403/)
Cardoso et al (https://link.springer.com/article/10.1007/s00203-021-02620-z)
Stopnisek and Shade (https://www.nature.com/articles/s41396-021-00955-5)
4. The majority of the data is shown in tables which do not provide good overviews of the results. For example number of taxa common to the different treatments can be shown in Venn diagrams and most abundant genera can be indicated in the text. The complete list is shown anyways in the supplementary tables. This would make the manuscript more appealing for the reader. For examples on how to display the data see Hakim et al (Point 6) and Xia et al (Point 1).
5. Some conclusions are not supported by the data. For example:
i. The authors claim that diversity is higher in Phaseolus, but they do not quantify it. They should include alfa-diversity analysis for all datasets to support their claims.
ii. The authors claim that co-inoculation with Azospirillum reduces nodule occupancy by Bradyrhizobium in soybean. This is based in relative abundances (RA). RA cannot be used to conclude this as for instance an increase of a different taxa (e.g. Methylobacterium) could lead to a reduction in the RA of Bradyrhizobium, even when absolute values are maintained. Non-rhizobial endophytes and rhizobia do not necessarily occupy the same space, as endophytes can colonise the apoplast of nodules while rhizobia infect nodules intracelullarly. To do statements about nodule occupancy, the authors must measure absolute abundance, either by using internal references during microbiome sequencing or by using qPCR. Otherwise, they need to modify the statement.
iii. Lines 138-140: “a smaller percentage of Bradyrhizobium was observed in the non-inoculated control treatment, on average 7.19 %, indicating the presence of several other microorganisms within that microbiome”. How do they conclude this from the smaller percentage? It is possible that the reduction is caused by the increase of a single type of microbe.
6. In the discussion the authors only vaguely compare their results with published data about the bean and soybean microbiomes (See point 3) and regarding nodule endophytes:
Mayhood and Mirza (https://www.ncbi.nlm.nih.gov/pmc/articles/PMC8117765/pdf/AEM.02884-20.pdf)
Crosbie et al (https://nph.onlinelibrary.wiley.com/doi/10.1111/nph.17988)
Hakim et al (https://www.sciencedirect.com/science/article/pii/S094450131930970X?via%3Dihub)
They should go beyond naming which taxa have been identified, but what are the key similarities and differences and how this connects to potential functions for selected taxa.
7. The authors should conduct rarefaction analyses and Beta diversity analyses.
Specific comments:
1. Revise title: the experimental design does not allow to conclude anything about the efficiency of BNF, remove second part of the title.
2. There are serious conceptual errors in the manuscript. For example in lined 73-75 the authors write: “Plants have microenvironment, called microbiomes…”. There are two generally accepted definitions for the microbiome, either all microorganisms in a particular environment or the combined genetic material of those microbes. The microbiome is not the environment.
3. Please clarify the following statements:
i. Lines 78-79: “…bacterial microbiomes associated with agricultural sustainability have attracted more attention than other groups of microorganisms”. How this connects to the previous sentence? This is a vague statement. What other groups of microorganisms?
ii. Lines 233-234: “… root microbiome of common bean showed higher variability (Table 6, Figure 1), confirming a more heterogenous environment than the nodule microbiome”. What do the authors mean with heterogeneous environment? Not clear why do they assume that a root is a more heterogeneous environment than a nodule. Do they refer to the physical/chemical environment within the different organs? Or to the microbiome, which is not an environment.
iii. Lines 321: “…additional high microbial community”. What do the authors mean? That the community is diverse, that specific members are abundant? If they mean diverse, relative to what? Nodules are incredibly restrictive and are colonised by an exquisitely narrow range of bacteria compared to roots or the rhizosphere.
iv. Lines 325: “… outstanding diversity”. How did you quantify diversity? Why is this outstanding? Outstanding relative to what?
v. Lines 356-358: “The inconsistent presence of some clades might indicate that they are not preferentially selected as endophytes, or might play limited plant-growth properties”. What do they mean with preferentially selected? What do they mean with play properties?
vi. Lines 394-395: “ …bacteria need to be able to adhere and proliferate on the root surface and penetrate cell walls without being tied down by the plant's immune system”. What bacteria? Most commensals are epiphytic or inhabit the apoplast. Thus there is no need to penetrate cell walls. If the authors mean rhizobia, they should specify it. But if this is the case I do not understand the connection to the previous sentence.
vii. Lines 414-415: “may unbalance the natural microbial community in root microbiomes”. Do they mean that inoculation can cause dysbiosis or that it can cause shifts in the natural microbial communities inhabiting different root compartments?
4. Binomial species names should be italicised. Examples: Lines 101, 107, 108, etc.
5. Figure 1 does not fit into page margins.
6. Figure 1 legend: what do the authors mean with “ of the three replicates”? Do they mean the three treatments? What does R1-R6 stand for? Replicates? Roots?
7. Reference for statement in lines 392-394 is missing.
8. 16S rRNA gene instead of 16S.
9. Line 541: DNeasy not DNasy.
10. Line 542: QIAGEN not Quiagen.
11. Line 558: PCR not PRC.
Author Response
In the manuscript entitled “Microbiome of Nodules and Roots of Soybean and Common Bean: Searching for Differences Associated with Contrasting Responses to Inoculation and Efficiency of Biological Nitrogen Fixation” the authors compare the microbiomes of soybean and common bean roots and nodules after different treatments (no inoculation, inoculation with rhizobia, co-inoculation with rhizobia and Azospirillum). Soybean and common bean are both important crops, but they differ in their promiscuity and their capacity to fix nitrogen symbiotically and thus their requirements for additional fertilisation. The authors report that in general the bacterial diversity in roots is higher than in nodules and that common bean nodules have a higher diversity than soybean nodules. They also report that the relative abundance of Bradyrhizobia changes upon co-inoculation with Azospirillum. This work adds to our understanding about how inoculation treatments impact microbial communities. In particular, how inoculation with elite rhizobia and Azospirillum, a widespread agricultural practice in Brazil one of the major soybean producers in the world, could impact root microbiomes.
Overall the design of the sampling is sound, and the description of the methods is detailed. However, the data generated is only superficially analysed and some conclusions are not supported by the data. Writing and data display should be improved. Examples illustrating these points are listed below.
General comments:
- The motivation for this work is not clearly defined. The authors try to address two loosely related questions:
- A) How the nodule microbiome affects symbiotic performance?
- B) What is the effect of different inoculation treatments on the root and nodule microbiomes?
Although Question A is in my opinion an extremely important question in the field of BNF, it cannot be answered using the experimental design presented here. Soybean and common bean are two distinct crops that were grown under different conditions (field vs. Greenhouse). Conclusions about the relationship between the microbiome and the symbiotic performance cannot be made.That is why in my opinion the authors should focus on question B. I recommend to re-write the paper to make this research question and its novelty clearer.
Reply: We clarified the points raised in the text. As explained to the other reviewer, it is still a paradox that we have two contrasting symbioses as those with the common bean and the soybean. And common bean almost always does not respond to inoculation under non-sterile conditions. We were the first to report, under non-sterile conditions, that inoculation with Bradyrhizobium increased their abundance in nodule microbiome, while in common bean the inoculation did not increase the Rhizobium abundance in the nodule microbiome. This result can be highly related to the lack of response to inoculation of common bean, contrarily to the soybean. In relation to the different conditions, we used non-sterile soil from the field to perform the greenhouse experiments. In the past few years, we have had serious pest and diseases problems with field-grown common bean. Therefore, there is need of strong application of fungicides and insecticides, impacting nodulation. To avoid this, we preferred to conduct the experiment under greenhouse conditions where we have no problems of pests and diseases and would have less impact in nodule and root microbiomes. We added this information in the material and methods section.
- The writing of the manuscript is not clear. Sentences are often too long and difficult to understand. For more specific examples see specific comments (Point 3).
Reply: We reviewed the manuscript and do believe that we made it clearer.
- Almost a third (25 out of 77) of the citations used in the manuscript and more that half (13 out of 23) of the citations used in the introduction are authored by the senior author. Important papers in the field are left out. For example, in the introduction (Lines 83-85) the authors state “…knowledge about the interactions of bacteria in the rhizosphere and inside the nodules is still scarce”. However there is a growing number of papers describing the root and nodule microbiomes, even some investigate the interaction between symbiotic performance and the nodule microbiome and the role of non-rhizobial endophytes. See more examples in point 5.
Reply: In the introduction, we clarified that the knowledge is still scarce when compared to the hundreds of studies isolating single microsymbionts from nodules. We do believe that maybe it was not clear enough that we were studying and discussing about nodule microbiome, not nodule functioning. Introduction should be not very long, therefore, in the discussion, we included the references recommended and we believe that the discussion was improved.
- Soybean
Cui et al (https://journals.asm.org/doi/epub/10.1128/mSystems.01299-20)
Reply: Included in the discussion
- Bean
Medina-Paz et al (https://www.ncbi.nlm.nih.gov/pmc/articles/PMC9269403/)
Reply: It was originally included in our manuscript. It was reference 54. Now we used more the results in the discussion. The study compared rhizosphere and endosphere microbiota, and in different stages of plant growth. Endosphere, composed by seeds and roots showed lower diversity in comparison to the rhizosphere. Therefore, we included in the discussion stating that from the rhizosphere to the nodules the specificity in the communities selected increased.
Cardoso et al (https://link.springer.com/article/10.1007/s00203-021-02620-z)
Reply: The study was focused more on different genotypes, biofortified or not, comparing plant rhizospheres and control bulk soils. It did not deal with symbiotic aspects, and therefore we did not include in the discussion.
Stopnisek and Shade (https://www.nature.com/articles/s41396-021-00955-5)
Reply: We included in our discussion. It is a very interesting manuscript, but dealing with core microbiomes of common bean rhizosphere, not roots or nodules. But we included in the discussion of some of the genera found, that were confirmed in our study, as well as the discussion about unknown genera.
- The majority of the data is shown in tables which do not provide good overviews of the results. For example number of taxa common to the different treatments can be shown in Venn diagrams and most abundant genera can be indicated in the text. The complete list is shown anyways in the supplementary tables. This would make the manuscript more appealing for the reader. For examples on how to display the data see Hakim et al (Point 6) and Xia et al (Point 1).
Reply: We changed all tables to figures, as requested. Thanks a lot for the suggestion, it is really more appealing now. To make the figures we had to re-analyze all data, and some taxonomic indications changed from December 2021 to now. We updated everything. We deleted all Tables with names of genera. Now everything is on Figures.
- Some conclusions are not supported by the data. For example:
- The authors claim that diversity is higher in Phaseolus, but they do not quantify it. They should include alfa-diversity analysis for all datasets to support their claims.
Reply: We were referring to detection of a higher number of genera, not diversity considering to alfa-diversity. We explained that in the text. We deleted all mentions to diversity, leaving number of genera, relative abundance of genera.
- The authors claim that co-inoculation with Azospirillum reduces nodule occupancy by Bradyrhizobium in soybean. This is based in relative abundances (RA). RA cannot be used to conclude this as for instance an increase of a different taxa (e.g. Methylobacterium) could lead to a reduction in the RA of Bradyrhizobium, even when absolute values are maintained. Non-rhizobial endophytes and rhizobia do not necessarily occupy the same space, as endophytes can colonise the apoplast of nodules while rhizobia infect nodules intracelullarly. To do statements about nodule occupancy, the authors must measure absolute abundance, either by using internal references during microbiome sequencing or by using qPCR. Otherwise, they need to modify the statement.
Reply: We included in the discussion the important observation that the endophytes can colonize the apoplast of nodules, while rhizobia infect nodules intracellularly. Interesting observation that we have forgotten to include. But we point out that the relative abundance of rhizobia was decreased, indicating that it could be a limitation to symbiotic performance, as they could be competing for nutrient and energy sources. Therefore, to be advantageous do plant growth, they must provide benefits that we still do not know, We comment that the results call the attention for the need of several other studies to clarify the role of these endophytes.
iii. Lines 138-140: “a smaller percentage of Bradyrhizobium was observed in the non-inoculated control treatment, on average 7.19 %, indicating the presence of several other microorganisms within that microbiome”. How do they conclude this from the smaller percentage? It is possible that the reduction is caused by the increase of a single type of microbe.
Reply: We re-phrased to delete the several and to state other microorganism or microorganisms.
- In the discussion the authors only vaguely compare their results with published data about the bean and soybean microbiomes (See point 3) and regarding nodule endophytes:
Reply: We included and discussed the references listed in point 3.
Mayhood and Mirza (https://www.ncbi.nlm.nih.gov/pmc/articles/PMC8117765/pdf/AEM.02884-20.pdf)
Reply: It was already included in the manuscript. It was citation 32.
Crosbie et al (https://nph.onlinelibrary.wiley.com/doi/10.1111/nph.17988)
Reply: Reference included and discussed.
Hakim et al
(https://www.sciencedirect.com/science/article/pii/S094450131930970X?via%3Dihub)
Reply: This reference was included before. Original reference 65.
They should go beyond naming which taxa have been identified, but what are the key similarities and differences and how this connects to potential functions for selected taxa.
Reply: In the discussion we included some putative functions.
- The authors should conduct rarefaction analyses and Beta diversity analyses.
Reply: As we explained in question 5, we were referring to detection of a higher number of genera, not diversity referring to alfa-diversity. We explained that in the text. We deleted all mentions to diversity, leaving only number and relative abundance of genera.
Specific comments:
- Revise title: the experimental design does not allow to conclude anything about the efficiency of BNF, remove second part of the title.
Reply: We changed the title.
- There are serious conceptual errors in the manuscript. For example in lined 73-75 the authors write: “Plants have microenvironment, called microbiomes…”. There are two generally accepted definitions for the microbiome, either all microorganisms in a particular environment or the combined genetic material of those microbes. The microbiome is not the environment.
Reply: Sorry for the mistake. It was a problem of writing. It is now correct.
- Please clarify the following statements:
- Lines 78-79: “…bacterial microbiomes associated with agricultural sustainability have attracted more attention than other groups of microorganisms”. How this connects to the previous sentence? This is a vague statement. What other groups of microorganisms?
Reply: Again, sorry. It was a problem of writing. Changed
- Lines 233-234: “… root microbiome of common bean showed higher variability (Table 6, Figure 1), confirming a more heterogenous environment than the nodule microbiome”. What do the authors mean with heterogeneous environment? Not clear why do they assume that a root is a more heterogeneous environment than a nodule. Do they refer to the physical/chemical environment within the different organs? Or to the microbiome, which is not an environment.
Reply: We corrected to clarify the sentence. We were referring to lower abundance of the specific microsymbiont.
iii. Lines 321: “…additional high microbial community”. What do the authors mean? That the community is diverse, that specific members are abundant? If they mean diverse, relative to what? Nodules are incredibly restrictive and are colonised by an exquisitely narrow range of bacteria compared to roots or the rhizosphere.
Reply: We changed to several other genera of bacteria.
- Lines 325: “… outstanding diversity”. How did you quantify diversity? Why is this outstanding? Outstanding relative to what?
Reply: Changed to outstanding number of genera.
- Lines 356-358: “The inconsistent presence of some clades might indicate that they are not preferentially selected as endophytes, or might play limited plant-growth properties”. What do they mean with preferentially selected? What do they mean with play properties?
Reply: We re-phrased stating that they are not usual endophytes.
- Lines 394-395: “ …bacteria need to be able to adhere and proliferate on the root surface and penetrate cell walls without being tied down by the plant's immune system”. What bacteria? Most commensals are epiphytic or inhabit the apoplast. Thus there is no need to penetrate cell walls. If the authors mean rhizobia, they should specify it. But if this is the case I do not understand the connection to the previous sentence.
Reply: We re-phrased to indicate that we were mentioning the bacteria that compose plant´s microbiome,
vii. Lines 414-415: “may unbalance the natural microbial community in root microbiomes”. Do they mean that inoculation can cause dysbiosis or that it can cause shifts in the natural microbial communities inhabiting different root compartments?
Reply: We-rephrased to cause shifts in the natural microbial communities and in root microbiomes
- Binomial species names should be italicised. Examples: Lines 101, 107,
108, etc.
Reply: Sorry, it was a problem in the process of transference to the journal´s file.
- Figure 1 does not fit into page margins.
Reply: We changed the layout. Now it is better to visualize the genera.
- Figure 1 legend: what do the authors mean with “ of the three replicates”? Do they mean the three treatments? What does R1-R6 stand for? Replicates? Roots?
Reply: Sorry, for the mistake, instead of three replicates was three treatments. We re-phrased and included that R1 to R6 refer to six biological replicates.
- Reference for statement in lines 392-394 is missing.
Reply: We were referring to our study. We clarified that.
- 16S rRNA gene instead of 16S.
Reply: Corrected.
- Line 541: DNeasy not DNasy.
Reply: Sorry, corrected now.
- Line 542: QIAGEN not Quiagen.
Reply: Sorry, corrected now.
- Line 558: PCR not PRC.
Reply: Sorry, corrected now.
Round 2
Reviewer 2 Report
The authors made some improvement of the manuscript, but the analysis of microbiome was not satisfied. Only relative abundance is not shown the real change of microbiome. How about the difference of total bacteria number?
Major revision
-There has no more analysis about bioinformatic data, such as α, β diversity or PCoA /NMDS.
Reply: We explained and changed throughout the manuscript to clarify that we were not performing a study about diversity in the strictu senso. Our main interest was to identify rhizobia and other genera in nodule and root microbiomes of the two contrasting symbioses, and to verify if there were shifts with inoculation. We believe that it is now clearer.
Q: All your conclusion are based on microbiome analysis, but I don’t think the authors made a professional microbiome analysis. The authors just showed the change of microbiome after inoculation, it is obviously. If the temperature is changes, I think the microbiome will be changed. But the deep analysis about why inoculation made the change of microbiome is no more data.
-There has no physiological analysis about BNF, Shoot biomass of each treatment.
Reply: Our study was performed based on the results reported in dozens of experiments performed for decades. The differences between the two symbioses are well known. Including two important recent reviews, cited in the manuscript (Peoples et al., 2021; Herridge et al., 2022). Therefore, we wanted to focus on the microbiomes. In the discussion, when we mention the differences, we cite five papers for the soybean and eight for the common bean.
Q: Microbiome did not give the answer why co-inoculation will increase symbiotic phenotype. The authors need isolate the typical bacteria from each treatment, and identify which one is the most important strain.
-There has no compatible relationship between soybean and common bean, for the authors inoculated different rhizobium on these two plants. I suggest the authors may inoculate one rhizobium which can form nodules both on soybean and common bean, such as NGR234.
Reply: NGR234 is not a good strain neither for common bean, nor for soybean. It nodulates the legumes, but it is not very efficient. We used the best elite strains, with known superior performance worldwide. Rhizobium tropici CIAT 899 is used in commercial inoculants for common beans at least in South America and Africa. And the two Bradyrhizobium strains are used in more than 100 million doses in Brazil, Bolivia, Paraguay.
Q: The authors did not understand what I mean. If you inoculated two different rhizobia on different legumes, how to compare? What is the CK? Soybean can compared each other, but I don’t think you can compare soybean with common bean.
-Why the authors inoculated two Bradyrhizobium and two A. brasilense?
Reply: Because that´s how they are used in the commercial inoculants in Brazil. We included this explanation in the results and in the material and methods section.
Q: I know, but why the authors only inoculated one type rhizobium on common bean? I curious about the design of experiments.
Author Response
The authors made some improvement of the manuscript, but the analysis of microbiome was not satisfied. Only relative abundance is not shown the real change of microbiome. How about the difference of total bacteria number?
Major revision
-There has no more analysis about bioinformatic data, such as α, β diversity or PCoA /NMDS.
Reply: We explained and changed throughout the manuscript to clarify that we were not performing a study about diversity in the strictu senso. Our main interest was to identify rhizobia and other genera in nodule and root microbiomes of the two contrasting symbioses, and to verify if there were shifts with inoculation. We believe that it is now clearer.
Q: All your conclusion are based on microbiome analysis, but I don’t think the authors made a professional microbiome analysis. The authors just showed the change of microbiome after inoculation, it is obviously. If the temperature is changes, I think the microbiome will be changed. But the deep analysis about why inoculation made the change of microbiome is no more data.
R2: We performed new analyses and included diversity data, alfa and beta-diversity. We do believe that the manuscript was enriched with the new data added.
-There has no physiological analysis about BNF, Shoot biomass of each treatment.
Reply: Our study was performed based on the results reported in dozens of experiments performed for decades. The differences between the two symbioses are well known. Including two important recent reviews, cited in the manuscript (Peoples et al., 2021; Herridge et al., 2022). Therefore, we wanted to focus on the microbiomes. In the discussion, when we mention the differences, we cite five papers for the soybean and eight for the common bean.
Q: Microbiome did not give the answer why co-inoculation will increase symbiotic phenotype. The authors need isolate the typical bacteria from each treatment, and identify which one is the most important strain.
R2: The strains used in the experiments derived from dozens of experiments and are listed as those recommended for use in commercial inoculants in Brazil. All commercial inoculants can use exclusively the strains listed as the most suitable for each crop. Last year over 100 million doses of inoculants were sold in the country. The strains used in our study are the best and came from years and years of selection and identification of elite strains. We used the best strains to achieve the best performance. Although we thought that this was clear in the manuscript, we reinforced the statement to make it more clear why we used these strains and that they are the best ones available for inoculation in Brazil. We also reinforced the explanation that commercial inoculants in Brazil carry two strains of Bradyrhizobium spp., while those for common bean carry only one strain of the Rhizobium tropici group.
-There has no compatible relationship between soybean and common bean, for the authors inoculated different rhizobium on these two plants. I suggest the authors may inoculate one rhizobium which can form nodules both on soybean and common bean, such as NGR234.
Reply: NGR234 is not a good strain neither for common bean, nor for soybean. It nodulates the legumes, but it is not very efficient. We used the best elite strains, with known superior performance worldwide. Rhizobium tropici CIAT 899 is used in commercial inoculants for common beans at least in South America and Africa. And the two Bradyrhizobium strains are used in more than 100 million doses in Brazil, Bolivia, Paraguay.
Q: The authors did not understand what I mean. If you inoculated two different rhizobia on different legumes, how to compare? What is the CK? Soybean can compared each other, but I don’t think you can compare soybean with common bean.
R2: We wanted to compare why one symbiosis is very specific and effective and the other one is very promiscuous and ineffective. Even when we use the best strains for each legume. The question that has been made for years and years is why soybean can supply almost 100% of host´s need with the contribution of BNF and common bean on average only 38%. That was the question. We tried to make it more clear now. For example, in the introduction, we write that “Interestingly, not all symbioses are equally effective in fixing N2, and understanding these differences represents a major challenge that can help designing strategies to increase the contribution of BNF”. The purpose of our study was to try to find some events that could help to explain the differences between the two symbioses. Still in the introduction, we explain the differences between soybean and common bean from lines 51 to 64. We attribute to some of the findings in our study at least putative explanations for the differences.
-Why the authors inoculated two Bradyrhizobium and two A. brasilense?
Reply: Because that´s how they are used in the commercial inoculants in Brazil. We included this explanation in the results and in the material and methods section.
Q: I know, but why the authors only inoculated one type rhizobium on common bean? I curious about the design of experiments.
R2: Because for the soybean crop commercial inoculants in Brazil carry those two strains, as the performance of two strains is better than one strain. The same for Azospirillum brasilense, they carry two strains. Contrarily, for common bean the best performance has been achieved with only one strain belonging to the R. tropici group. We tried to make this point clearer in the manuscript. We emphasized this also at the material and methods section.
Reviewer 3 Report
The authors addressed the points that I raised in the first revision, however not completely to my satisfaction. Below is a detailed list with points that were not improved.
1. The authors changed the title, however they do not really address my main criticism that is that with their experimental design they cannot compare nitrogen fixation differences between soybean and bean. I understand that there are experimental limitations to field experiments, but we cannot force conclusions due to necessity. The authors could have grown both plants under green-house conditions using the same soil. Now how can the authors be sure that the differences that they observe are due to the host (genotype, promiscuity, etc) or due to the different soils, temperature, humidity, etc. Soil has been shown to be the major determinant for nodule microbiota. In addition, the authors do not do any measurements of nitrogen fixation, plant growth, etc. I understand that this has been described in the literature, but you cannot extrapolate data from other studies.
Moreover, in the manuscript the authors do not really compare the results obtained for both hosts. They describe the results for soybean and then they describe the results for bean. Then why in the title do they emphasise the comparison?
I think the data is still valuable, simply don’t overstate it.
2. The writing is better now, but there are still parts that are difficult to read and there are multiple typos. I recommend to ask a native speaker or a professional editing service to proof-read the manuscript. Examples Line 350 (level not leve), Line 667 (combinations not combiantions), Line 679 (sterile not setrile), Line 708-709 (surface sterilised not sterilisation, sterile water not sterilised), Line 710 (Plant tissues were divided into, not plants), Line 714 (repetition of line 711), and many more.
3. It is not clear to me why the authors do not conduct diversity analyses with their data. It is a lost opportunity. The power of microbiome research is in the statistical analyses that can be done. It allows to go beyond listing taxa.
4. The authors should indicate if the genera that they list were present in all replicates. They could add supplementary tables indicating the abundance in the different replicates as they did for the Unc taxa.
5. The authors do not convincingly address point 5.ii. Why reduced RA would indicate a limitation to symbiotic performance. Reduced RA does not mean reduced absolute abundance. You can still have the same number of rhizobia but have a reduced RA (see explanation in previous revision). If you have the same number of rhizobia why would symbiotic performance be limited. Please make clear in the text that this is also a possibility.
In their reply the authors speculate that they rhizobia and endophytes could compete for nutrients and energy resources. To my knowledge there are no studies about the metabolism of endophytes compared to rhizobia, they could feed on completely different carbon sources for example. In lines 456-469 the authors attribute the RA reduction to competitiveness in the rhizosphere. This makes me believe that the authors still think that a reduction in RA means a reduction in nodule colonisation. In addition, why competition in the rhizosphere would affect absolute abundance in nodules. A nodule can be fully colonised after a single microcolony event. I understand that competition could lead to a reduced number of nodules, but once the nodule is formed, why would it not be fully colonised by the bacteria entrapped in the microcolony?
6. There are still conceptual mistakes in the text. For example, in the previous revision (Specific comment point 2) I pointed out problems with the concept of microbiome. The authors modified the sentence, but it is still wrong. Lines 76-77: microbiomes are not physical spaces, so they cannot “house” communities.
7. Make sure that text in the figures is readable and that figures fit within the page margins.
8. Not clear why some taxa changed compared to the previous version. Were there more analyses conducted? If yes, what was different?
9. When the authors describe the results in the figures they select some taxa as examples, not clear if there are the most abundant, the most prevalent among different replicates or if they were randomly selected.
Author Response
Comments and Suggestions for Authors
The authors addressed the points that I raised in the first revision, however not completely to my satisfaction. Below is a detailed list with points that were not improved.
- The authors changed the title, however they do not really address my main criticism that is that with their experimental design they cannot compare nitrogen fixation differences between soybean and bean. I understand that there are experimental limitations to field experiments, but we cannot force conclusions due to necessity. The authors could have grown both plants under green-house conditions using the same soil. Now how can the authors be sure that the differences that they observe are due to the host (genotype, promiscuity, etc) or due to the different soils, temperature, humidity, etc. Soil has been shown to be the major determinant for nodule microbiota. In addition, the authors do not do any measurements of nitrogen fixation, plant growth, etc. I understand that this has been described in the literature, but you cannot extrapolate data from other studies.
Moreover, in the manuscript the authors do not really compare the results obtained for both hosts. They describe the results for soybean and then they describe the results for bean. Then why in the title do they emphasise the comparison?
I think the data is still valuable, simply don’t overstate it.
R2: As we commented before, it calls the attention why both symbioses are so different in symbiotic performance. Any information about this is very important to try to improve BNF contribution. We tried to make clearer now in the discussion, stating that this is only one step that might contribute to the understanding. But we added more differences between the two symbiosis, by analyzing diversity indices, not evaluated in the previous version.
- The writing is better now, but there are still parts that are difficult to read and there are multiple typos. I recommend to ask a native speaker or a professional editing service to proof-read the manuscript. Examples Line 350 (level not leve), Line 667 (combinations not combiantions), Line 679 (sterile not setrile), Line 708-709 (surface sterilised not sterilisation, sterile water not sterilised), Line 710 (Plant tissues were divided into, not plants), Line 714 (repetition of line 711), and many more.
R2: Sorry for the mistakes. We had some difficulties when transposing to the template. We verified all the manuscript for misspelling and grammar mistakes.
- It is not clear to me why the authors do not conduct diversity analyses with their data. It is a lost opportunity. The power of microbiome research is in the statistical analyses that can be done. It allows to go beyond listing taxa.
R2: We worked very hard again in all data and performed new analyses for diversity. The manuscript includes now diversity data.
- The authors should indicate if the genera that they list were present in all replicates. They could add supplementary tables indicating the abundance in the different replicates as they did for the Unc taxa.
R2: In the first version we had all tables. However, one reviewer asked to transform the tables in figures. Now we have returned the tables as supplementary material, as they indicate the percentages, and continued with the figures, more didatic. We have now ten Supplementary Tables.
- The authors do not convincingly address point 5.ii. Why reduced RA would indicate a limitation to symbiotic performance. Reduced RA does not mean reduced absolute abundance. You can still have the same number of rhizobia but have a reduced RA (see explanation in previous revision). If you have the same number of rhizobia why would symbiotic performance be limited. Please make clear in the text that this is also a possibility.
In their reply the authors speculate that they rhizobia and endophytes could compete for nutrients and energy resources. To my knowledge there are no studies about the metabolism of endophytes compared to rhizobia, they could feed on completely different carbon sources for example. In lines 456-469 the authors attribute the RA reduction to competitiveness in the rhizosphere. This makes me believe that the authors still think that a reduction in RA means a reduction in nodule colonisation. In addition, why competition in the rhizosphere would affect absolute abundance in nodules. A nodule can be fully colonised after a single microcolony event. I understand that competition could lead to a reduced number of nodules, but once the nodule is formed, why would it not be fully colonised by the bacteria entrapped in the microcolony?
R2: First, we corrected all manuscript to make it clear that we were referring to relative abundance, not only abundance. In relation to the question of why reduce RA would affect symbiotic performance, we refer to it as a possibility. The hypothesis is now stronger with the results obtained in the new diversity analyses. We are raising possibilities according to the results found. In relation to the nutrition, we know that nutrients are always limited by the plants. The nutrients required by an associative or endophyte is the same as a rhizobium, as demonstrated by the very similar amounts of nutrients in culture media. The endophytes can colonize the nodules in early steps of the infection process. Studies cited in the manuscript about vertical transference of microbiomes from seeds also indicate this.
- There are still conceptual mistakes in the text. For example, in the previous revision (Specific comment point 2) I pointed out problems with the concept of microbiome. The authors modified the sentence, but it is still wrong. Lines 76-77: microbiomes are not physical spaces, so they cannot “house” communities.
R2: Sorry, we thought that we left exactly as asked before. We probably did not understand what you have suggested. We corrected it again.
- Make sure that text in the figures is readable and that figures fit within the page margins.
R2: Yes, we checked the figures.
- Not clear why some taxa changed compared to the previous version. Were there more analyses conducted? If yes, what was different?
R2: Yes, we had to re-analyze all data again. The first analysis was finished by January and now we re-analyzed everything. Some slight changes occurred because taxonomy had changes in this period in the databases consulted.
- When the authors describe the results in the figures they select some taxa as examples, not clear if there are the most abundant, the most prevalent among different replicates or if they were randomly selected.
R2: We selected those with higher probability of interfering in plant growth according to results from the literature. We added a comment on that. We also discussed the genera that had been reported in previous studies. But all information is available for the readers, now including all data in supplementary material.